# Dissecting the pre-placodal transcriptome to reveal presumptive direct targets of Six1 and Eya1 in cranial placodes

**Nick Riddiford[1,2], Gerhard Schlosser[1,2]\***

[1]School of Natural Sciences, National University of Ireland, Galway, Ireland; [2]Regenerative Medicine Institute (REMEDI), National University of Ireland, Galway, Ireland

**Abstract** The pre-placodal ectoderm, marked by the expression of the transcription factor Six1 and its co-activator Eya1, develops into placodes and ultimately into many cranial sensory organs and ganglia. Using RNA-Seq in *Xenopus laevis* we screened for presumptive direct placodal target genes of Six1 and Eya1 by overexpressing hormone-inducible constructs of Six1 and Eya1 in pre-placodal explants, and blocking protein synthesis before hormone-inducing nuclear translocation of Six1 or Eya1. Comparing the transcriptome of explants with non-induced controls, we identified hundreds of novel Six1/Eya1 target genes with potentially important roles for placode development. Loss-of-function studies confirmed that target genes encoding known transcriptional regulators of progenitor fates (e.g. Sox2, Hes8) and neuronal/sensory differentiation (e.g. Ngn1, Atoh1, Pou4f1, Gfi1) require Six1 and Eya1 for their placodal expression. Our findings provide insights into the gene regulatory network regulating placodal neurogenesis downstream of Six1 and Eya1 suggesting new avenues of research into placode development and disease.

**\*For correspondence:** gerhard. schlosser@nuigalway.ie

**Competing interests:** The authors declare that no competing interests exist.

## Introduction

The cranial placodes give rise to many sense organs of the vertebrate head (including nose, ear and lateral line) and contribute to the anterior pituitary and sensory ganglia of the cranial nerves. Together with the neural crest, which also contributes to cranial ganglia as well as the head skeleton, they originated as an evolutionary novelty in stem vertebrates, on the adoption of a more active and exploratory life style (*Northcutt and Gans, 1983*; *Schlosser, 2015*). Defects in placode development underlie many congenital diseases of sensory organs and the endocrine system (*Petit et al., 2001*; *Davis et al., 2013*; *Xu, 2013*), however, despite this central importance of placodes in the evolution and development of the vertebrate head, they have been much less well studied than the neural crest, and little is known about the gene regulatory networks (GRNs) driving early placode development.

Fate mapping studies have shown that all cranial placodes develop from a common precursor region, the pre-placodal ectoderm (PPE) (*Streit, 2002*; *Bhattacharyya et al., 2004*; *Xu et al., 2008*; *Pieper et al., 2011*). In neural plate stage embryos, the PPE is located as a horseshoe-shaped domain around the anterior neural plate (and abutting the cranial neural crest laterally) which subsequently breaks up into individual placodes (*Schlosser, 2010*; *Grocott et al., 2012*; *Saint-Jeannet and Moody, 2014*). Molecularly, the PPE is characterised by the expression of *Six1* and *Eya1*, which also continues in most placodes derived from the PPE (*Schlosser and Ahrens, 2004*). Whereas *Six1* encodes a transcription factor, *Eya1* encodes a transcriptional co-activator that also has phosphatase activity (*Kumar, 2009*; *Tadjuidje and Hegde, 2013*), and Six1 and Eya1 have been shown to form a protein complex and synergistically activate transcription (*Ohto et al., 1999*;

**eLife digest** Animals that possess a backbone – also known as vertebrates – have several paired sense organs in their heads, such as the eyes and the olfactory system. These organs are thought to have arisen as vertebrates evolved from their filter-feeding ancestors and adopted an increasingly active and predatory lifestyle. The cranial placodes are tissues in the vertebrate embryo that give rise to many of these sense organs early in development. The sensory neurons that transmit information from the organs to the brain – including those that process hearing and smell – also develop from these tissues.

While much is known about how the sense organs work, relatively little is known about the early processes involved in their development. Two genes have been established as crucial for the formation and later development of the sense organs. These genes encode two proteins called Six1 and Eya1 that regulate other genes, although the identity of the genes that they target in placodes was not known.

Using computational and experimental approaches in the African clawed frog (*Xenopus laevis*), Riddiford and Schlosser identified hundreds of genes that are regulated by both Six1 and Eya1 in placodes. Many of these targeted genes regulate the activity of other genes and direct important cell decisions, such as whether a stem cell should develop into a neuron. After validating several of these targets in the laboratory, Riddiford and Schlosser proposed a network of gene regulation, activated by Six1 and Eya1, that drives sense organ development in vertebrates.

While Riddiford and Schlosser provide strong evidence that Six1 and Eya1 directly regulate many of the newly identified genes, further work is required to firmly establish this. Future studies could also explore how Six1 and Eya1 drive seemingly contradictory cellular decisions, encouraging some stem cells to develop into neurons and others to maintain a stem-like state.

*Li et al., 2013*). However, both Six1 and Eya1 also interact with other protein interaction partners; Six1, for example, has been shown to act as a transcriptional repressor after binding to the co-repressor Groucho (*Brugmann et al., 2004*) whereas Eya1 is known to form protein complexes with other binding partners including the transcription factor Sox2 (*Ahmed et al., 2012a*; *Tadjuidje and Hegde, 2013*).

Loss of Six1 or Eya1 function in mouse, zebrafish, chick or *Xenopus* embryos leads to a similar spectrum of PPE and placodal defects, with altered expression of other PPE genes, decreased proliferation and increased apoptosis in many placodes, compromised morphogenetic movements (invagination or cell delamination) and a decreased production of sensory cells and neurons (*Xu et al., 1999*; *Laclef et al., 2003*; *Zheng et al., 2003*; *Brugmann et al., 2004*; *Zou et al., 2004*; *Kozlowski et al., 2005*; *Schlosser et al., 2008*; *Christophorou et al., 2009*; *Ahmed et al., 2012a, 2012b*). In human patients, mutations in both Six1 and Eya1 lead to branchio-oto-renal (BOR) and branchio-otic (BO) syndromes with congenital hearing loss (*Kochhar et al., 2007*). These findings suggest that these proteins are core regulators of placode development and promote multiple aspects of placode development synergistically, although Eya1-independent roles of Six1 have also been reported (*Brugmann et al., 2004*; *Bricaud and Collazo, 2011*). Specifically, Six1 and Eya1 have been shown to play central roles, during multiple steps, in the development of sensory cells (e.g. hair cells in the inner ear) as well as sensory neurons, and promote both the proliferation of sensory/neuronal progenitors as well as sensory and neuronal differentiation in a dosage dependent fashion (*Zou et al., 2004*; *Schlosser et al., 2008*; *Zou et al., 2008*; *Ahmed et al., 2012b, 2012a*). Recently Atoh1, an essential determination gene for hair cell development, has been shown to be directly transcriptionally activated by Six1/Eya1 binding to its enhancer (*Ahmed et al., 2012a*). Moreover, the neuronal progenitor genes *Sox2* and *Sox3* have been shown to be up-regulated by Six1 and Eya1 in the absence of protein synthesis, suggesting that they are also direct target genes (*Schlosser et al., 2008*). Several other direct target genes of Six1 have been identified (*Kumar, 2009*; *Xu, 2013*), but no specific screen for direct target genes of Six1 and Eya1 in the PPE and the developing placodes has yet been conducted.

Here, using RNA-Seq in *Xenopus laevis*, we present the first comprehensive screen for presumptive direct target genes of Six1 and Eya1 in the developing placodes in any vertebrate. Hormone-inducible constructs of Six1 and Eya1 (fused with the human glucocorticoid receptor [GR]) were overexpressed, either alone or in combination, in *Xenopus* embryos. We then explanted the PPE at neural fold stages and activated nuclear translocation of Six1 or Eya1 in these explants by the addition of dexamethasone (DEX) after blocking protein synthesis by cycloheximide (CHX). This approach has previously been shown to reliably activate direct targets of GR-fusion constructs only in the presence of DEX (*Kolm and Sive, 1995*; *Seo et al., 2007*). We then analysed the transcriptome of placodal explants by RNA-Seq and compared this to control explants which were not hormone induced, in order to specifically survey target genes directly activated or repressed by Six1 or Eya1 in the PPE and developing placodes. Using this method, we were able to identify a large number of novel target genes with potentially important roles for cranial placode development. We were further able to show in loss of function studies that several target genes encoding known regulators of progenitor fates (e.g. *Sox2*, *Hes8*) and neuronal/sensory differentiation (e.g. *Ngn1*, *Atoh1*, *Pou4f1.2*, *Gfi1a*) required both Six1 and Eya1 for their expression in the developing placodes. Our findings provide pioneering insights into the GRNs regulating placode development downstream of Six1 and Eya1, and suggest exciting new avenues of research for understanding placode development and disease.

## Results

### The pre-placodal transcriptome

RNA was extracted from explants cut from the PPE of un-injected embryos and characterised using RNA-Seq to provide a complete transcriptome of the PPE. After removing genes expressed at low levels (FPKM < 1) and annotation against a *Xenopus* mRNA database (see Materials and methods), we assembled a transcriptome comprising 15,794 transcripts, and the top 1000 expressed genes are shown in *Supplementary file 1*. Gene Set Enrichment Analysis (GSEA) on these genes revealed that RNA processing/splicing was very highly enriched in the PPE transcriptome (enrichment score [E]: 43), suggesting that RNA-binding proteins and mRNA splicing mechanisms may play an important role in placodal development as has also been reported for the neural crest (*Simões-Costa et al., 2014*). Translation elongation and ribosomal proteins were also highly enriched (E: 32), perhaps reflecting the high rate of protein turnover in the rapidly changing PPE (*McCabe et al., 2004*).

### Identification of direct targets of Six1 and Eya1 in PPE

To identify presumptive direct targets of Six1 and Eya1, Six1-GR and Eya1-GR fusion proteins were overexpressed either alone or together in the PPE. In combination with a protein synthesis inhibitor (CHX), nuclear translocation of Six1 and Eya1 was induced by adding DEX for 2.5 hr, and gene expression was analysed using RNA-Seq (*Figure 1*). Presumptive direct targets of Six1 and Eya1 were determined by comparing Six1-GR-, Eya1-GR- or Six1-GR+Eya1-GR-injected embryos treated with CHX alone (as controls) against CHX+DEX-treated samples. Resultant data sets from such individual treatment groups (each with two biological replicates) are henceforth referred to as $Six1_i$, $Eya1_i$ and $Six1+Eya1_i$. In this paradigm, the expression of target genes for which either Six1 or Eya1 concentrations are limiting in the PPE should be affected in $Six1_i$ and $Eya1_i$ treatment groups, respectively (and potentially also in $Six1+Eya1_i$), while the expression of target genes limited by both Eya1 and Six1 concentrations in the PPE should be modulated only in the $Six1+Eya1_i$ treatment group.

Using this approach, we identified 365 genes up-regulated at least twofold that satisfied all criteria for differential expression (log$_2$ fold change [FC] $\geq$ 1; FPKM $\geq$ 1; FC < 0.5 in un-injected control) in $Six1_i$, 508 in $Eya1_i$ and 836 in $Six1+Eya1_i$ treatment groups, while 292 genes were down-regulated in $Six1_i$, 218 in $Eya1_i$ and 490 in $Six1+Eya1_i$ treatment groups (*Figure 2A and B*; *Supplementary file 2*). As an initial means of estimating data quality, we searched for targets of Six1 established in previous studies (*Atoh1* (*Ahmed et al., 2012a*); *Slc12a2* (*Ando et al., 2005*); *CyclinA1* (*Coletta et al., 2004*); *CyclinD1* (*Li et al., 2013*); *c-Myc* (*Li et al., 2003*); *Ezrin* (*Yu et al., 2006*); *Gdnf* (*Li et al., 2003*); *Sox3* (*Schlosser et al., 2008*); *Sox2* (*Schlosser et al., 2008*); *Sall1* (*Chai et al., 2006*); and *MyoD1* (*Liu et al., 2013*)) in the $Six1_i$ and $Six1+Eya1_i$ data sets. With the exception of *c-Myc,* all genes were present in the transcriptome, and most were found in either $Six1_i$ (*CyclinD1 [ccndx]* FC:

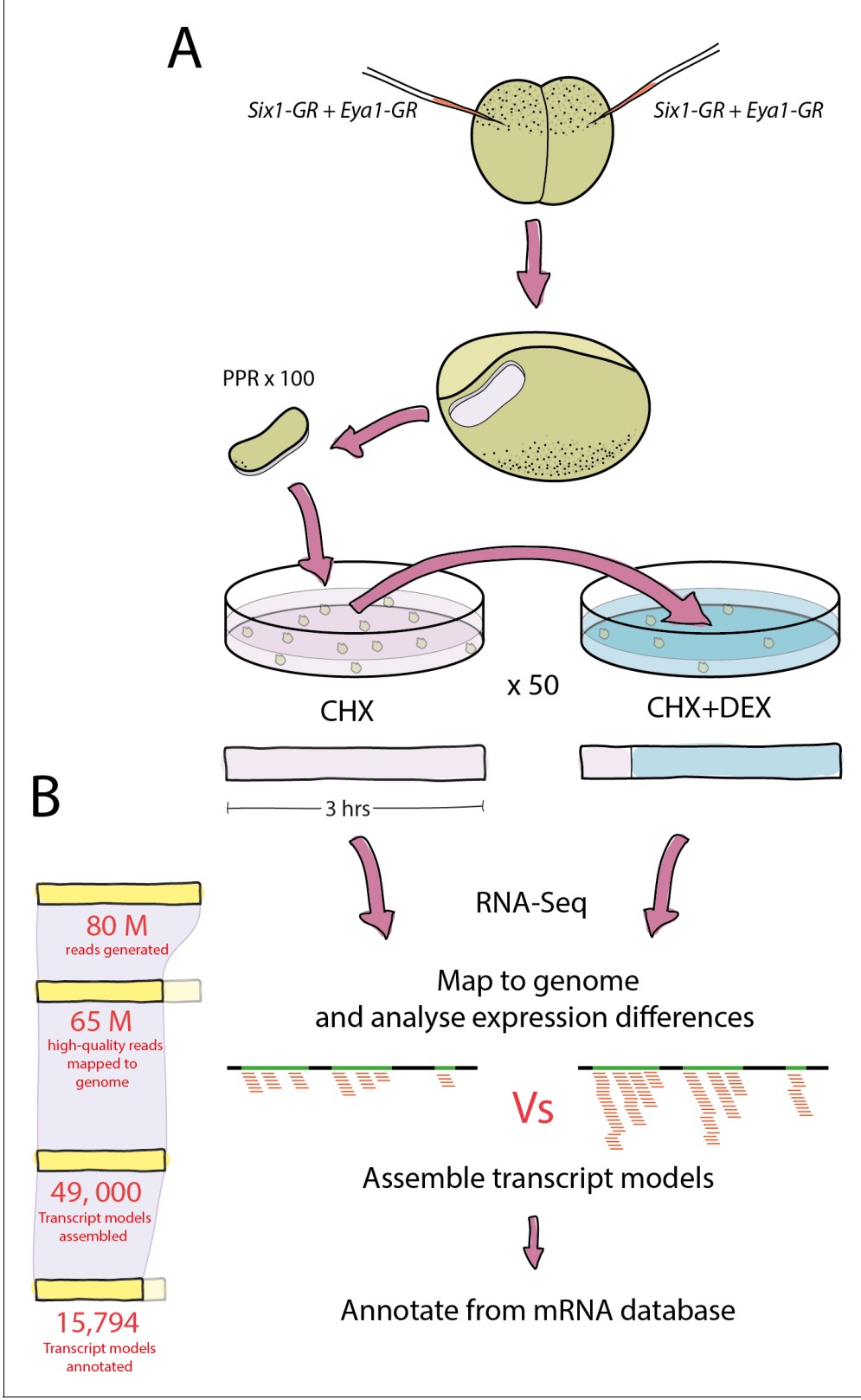

**Figure 1.** Experimental pipeline and overview of bioinformatic analysis. (**A**) Both blastomeres of two-cell stage embryos were injected with Six1-GR, Eya1-GR or Six1-GR+Eya1-GR and explants were cut from pre-placodal ectoderm. Explants were incubated in CHX for 30 min before being split into two groups; 50% were kept in CHX for 2.5 hr and 50% were transferred to CHX+DEX for 2.5 hr. RNA was extracted from both treatment groups and submitted to RNA-Sequencing. (**B**) On average, 80 million reads were generated in sequencing for each treatment group, and 65 million quality-

*Figure 1 continued on next page*

*Figure 1 continued*

trimmed reads were successfully mapped to the *Xenopus* genome. An average of 49,000 transcript models were assembled, of which 80% (39,000) were successfully annotated against a *Xenopus* mRNA database. Annotated transcript models were then filtered to condense duplicate annotations into 15,794 uniquely annotated transcript models, and differential expression analysis was then performed using CHX treated explants as a control for those treated with CHX+DEX.

7.48; *Slc12a2*, FC: -2.75; *CyclinA1*, FC: -3.68; *Sox2*, FC: 1.2; *MyoD, FC: 3.4)* or Six1+Eya1$_i$ (*Sox3*, FC: 0.9; *Atoh1*, FC: 1.4; *Sall1*, FC: 0.99) data sets, confirming the utility of our approach in identifying direct targets. Moreover, *Atoh1, Sox2* and *MyoD1* were found both in our Six1+Eya1$_i$ and Eya1$_i$ datasets as expected based on the known coregulation of these Six1 target genes by Eya1 (*Ahmed et al., 2012a*; *Grifone et al., 2007*; *Schlosser et al., 2008*). We suggest that overexpression of Eya1 alone may upregulate such genes in those parts of the ectoderm where *Six1* is already expressed at high levels but *Eya1* at relatively low levels in vivo.

## Six1 and Eya1 co-regulate many but not all PPE target genes

Comparison between our different treatment groups allows us to distinguish genes likely co-regulated by Six1 and Eya1 from those that are not and, thus, may be regulated by Six1 or Eya1 alone or in conjunction with other protein-binding partners. Since ectodermal expression of *Six1* and *Eya1* is widely overlapping in vivo but not completely congruent, genes co-regulated by Six1 and Eya1 may be differentially expressed not only after coinjection of Six1 and Eya1 (Six1+Eya1$_i$ treatment group) but also after injection of Six1 or Eya1 alone (Six1$_i$ and Eya1$_i$ treatment groups, respectively) because elevation of Six1 or Eya1 levels will produce higher levels of the coregulatory complex in those parts of the ectoderm where the respective protein is expressed at much lower levels than its binding partner. Hence, a subset of target genes with high response thresholds to the Six1-Eya1 coregulatory complex (e.g. due to low affinity binding sites) will respond to overexpression of Six1 or Eya1 alone with differential expression in these parts of the ectoderm while another subset of genes with low response thresholds (e.g. due to high affinity binding sites) will not. The latter subset will, thus, only be differentially expressed after overexpression of both Six1 and Eya1, creating expanded areas of Six1 and Eya1 coexpression in the ectoderm. Notably, the false discovery rate is expected to be lower for the former subset, which is supported by three independent treatment groups (Six1$_i$, Eya1$_i$ and Six1+Eya1$_i$), than in the latter subset, supported only by one (Six1+Eya1$_i$).

About half of all genes differentially expressed in the PPE in our various treatment groups show evidence of co-regulation by Six1 and Eya1. This includes 690 (633+57) up-regulated and 444 (440 +4) down-regulated genes (*Figure 2A and B*). Indeed, the top 10% of transcripts (ranked by FC; post DEX-filtering) up-regulated in Six1$_i$, Eya1$_i$ or Six+Eya1$_i$ treatment groups were each highly enriched for the top 10% of transcripts up-regulated in any of the other experimental treatment groups ($p<0.0001$; Fisher's exact test). More genes co-regulated by Six1 and Eya1 were up-regulated than were down-regulated (690/1134 = 60.8% for all co-regulated genes, 57/61 = 93.4% for co-regulated genes identified in each treatment group; *Figure 2A and B*), corroborating previous findings that Six1 and Eya1 typically act synergistically to activate transcription (*Ahmed et al., 2012b*, *2012a*; *Brugmann et al., 2004*; *Christophorou et al., 2009*; *Li et al., 2003*; *Ruf et al., 2004*). However, our identification of a subset of genes synergistically down-regulated by Six1 and Eya1 suggests that Eya1 may not always act as a co-activator of Six1.

In contrast, there is no support for co-regulation for genes that are differentially expressed only in Six1$_i$ but not Eya1$_i$ treatment groups (and vice versa) even for those genes that are also differentially expressed after Six1+Eya1$_i$ treatment. We identified 283 (190+93) genes up-regulated and 270 (233 +37) genes down-regulated by Six1 but not Eya1, indicating that these are regulated by Six1 in an Eya1 independent way but possibly dependent on other co-factors. Conversely, we identified 426 (373+53) genes up-regulated and 196 (187+9) genes down-regulated by Eya1 but not Six1 (*Figure 2A,B*) suggesting that these are regulated by Eya1 in conjunction with transcription factors other than Six1.

To add statistical power to our analysis, we next merged treatment groups and determined significantly differentially expressed genes ($q<0.05$) in these merged groups. We first created a data set Six1+Eya1$_m$ in which all replicates that involved overexpression of either Six1 or Eya1 were

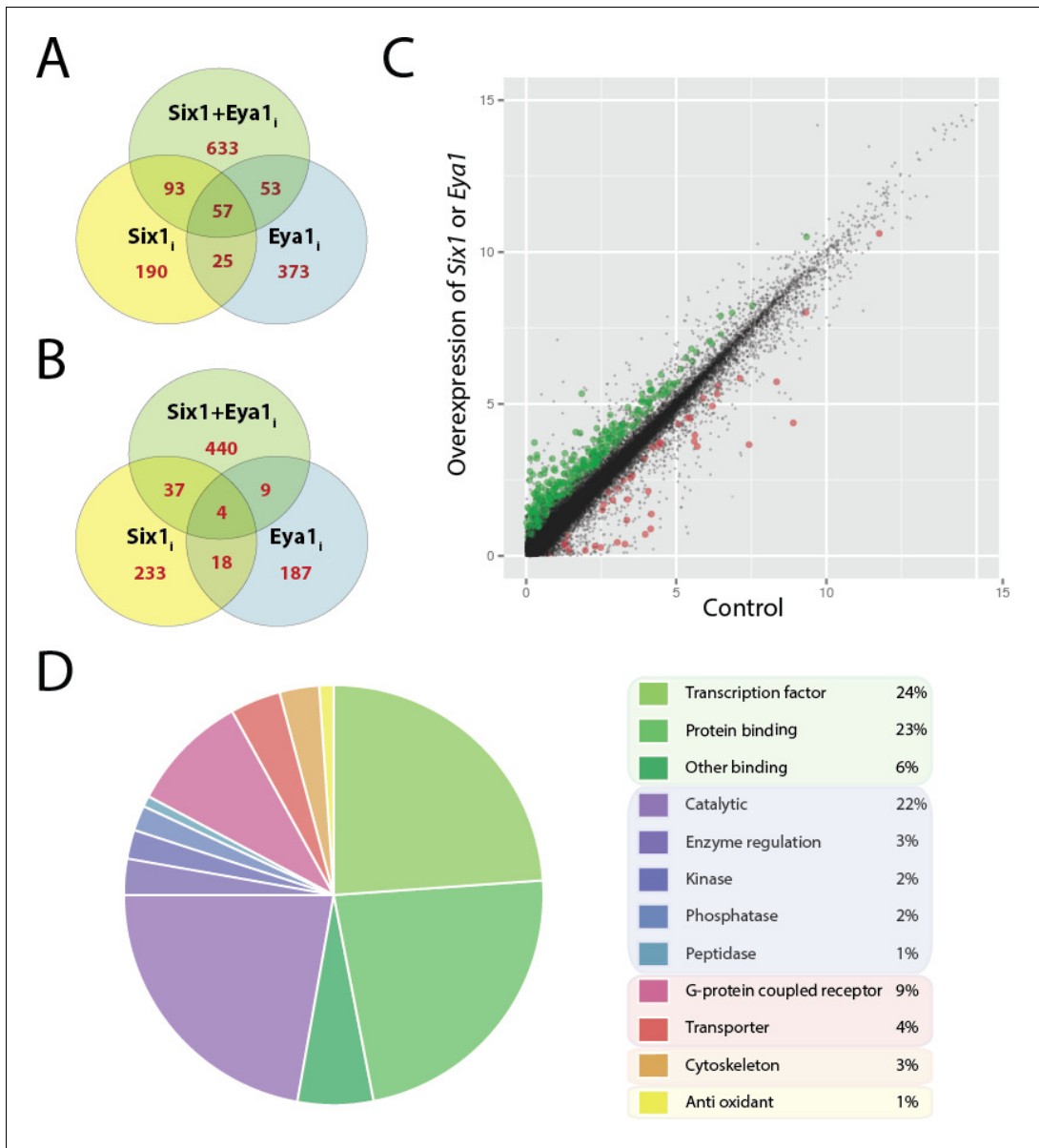

**Figure 2.** Differentially expressed genes after overexpression of *Six1* and *Eya1* in the PPE. Plots **A** and **B** show number of genes differentially regulated after overexpression of *Six1* alone (Six1$_i$; yellow), *Eya1* alone (Eya1$_i$; blue) or *Six1* and *Eya1* combined (Six1+Eya1$_i$; green). Each Venn diagram shows the number of genes (red) unique for each treatment group or shared between them. (**A**) Number of genes up-regulated and (**B**) down-regulated after injection with *Six1, Eya1* or *Six1+Eya1*. (**C**) The merged analysis resulted in hundreds of significantly differentially expressed genes in the PPE data set. Plot shows log$_2$ transformed (FPKM+1) values after overexpression of *Six1 or Eya1* (combination of all treatment groups; Six1+Eya1$_m$). Green points represent significantly (q<0.05) up-regulated genes and red points show significantly down-regulated genes. Plot **D** shows the enrichment of molecular function terms after overexpression of *Six1 or Eya1* based on significantly differentially expressed genes from the merged data set (Six1+Eya1$_m$; **Supplementary file 3,** Table 5). The area of the pie represents the total number of functional terms contained in the analysis, with each slice representing the percentage of genes against this total. Molecular functions shown can be broadly divided into five categories: Green slices are related to binding functions (53%); purple/blue represents enzyme activity (30%); pink/red shows transmembrane proteins (13%); orange cytoskeleton (3%) and yellow anti-oxidant (1%).

considered as equivalent (injection of Six1-GR, Eya1-GR or Six1-GR+Eya1-GR; 6 replicates in total). This allowed us to identify genes that are significantly differentially expressed across all treatment groups. We also created a data set Six1$_m$, in which all replicates that involved Six1 overexpression were considered as equivalent (injection of Six1-GR or Six1-GR+Eya1-GR; 4 replicates). This allowed us to identify genes with significant differential expression after Six1 upregulation. Similarly, we created data set Eya1$_m$ based on all replicates that involved Eya1 overexpression (injection of Eya1-GR or Six1-GR+Eya1-GR; 4 replicates) allowing us to identify genes differentially expressed after Eya1 upregulation. We found 181 significantly (q<0.05) up-regulated genes in the Six1+Eya1$_m$ group, 149 in Six1$_m$ and 112 in Eya1$_m$ (*Supplementary file 3*, Tables 1–3). Substantially fewer genes were negatively regulated in these merged groups, with only 14 significantly down-regulated genes found in Six1+Eya1$_m$, 11 in Six1$_m$ and 13 in Eya1$_m$ (*Supplementary file 3*, Tables 4–6), re-enforcing the notion that together, Six1 and Eya1 act primarily as transcriptional activators (*Figure 2C*).

## Target genes of Six1 and Eya1 are implicated in sensory neurogenesis

Presumptive direct targets that were significantly up-regulated in our merged data set (Six1+Eya1$_m$) were analysed using Panther (*Mi et al., 2013*) to examine the representation of genes grouped by molecular function (*Figure 2D*). Transcription factors and protein binding together accounted for the largest fraction of up-regulated genes (53% in total), followed by enzymes (30%) and transporter molecules (13%), suggesting a developmental function of many of the genes up-regulated by either Six1 or Eya1. GSEA was then conducted using DAVID (*Huang et al., 2009*) on the sets of significantly up- or down-regulated genes in our merged data sets, as well as in various combinations of subsets of differentially expressed genes from our individual treatment groups (*Figure 3* and *Figure 3—figure supplement 1*). This analysis showed that genes directly up-regulated by Six1, Eya1 or Six1+Eya1 were highly enriched for terms associated with sense organ development, inner-ear development, mechanoreceptor differentiation, eye morphogenesis, neurogenesis and axon guidance consistent with their synergistic role in sensory development (*Grocott et al., 2012*; *Schlosser, 2010*) and neurogenesis (*Maier et al., 2014*; *Schlosser and Northcutt, 2000*). Apart from genes encoding transcription factors involved in sensory development (see below), genes encoding cell cycle regulators (*CyclinD, RGCC*), cell surface receptors and adhesion molecules (e.g. *CXCR7, EDAR, Protocadherin11, Claudin3, Fzd1, Fzd4,*), secreted proteins (e.g. *FGF3, FGF19, Dkk1, Neurotrophin3*) and cytoskeletal regulators (e.g. *RhoV, Espin*) with known or potential roles in placode development were also up-regulated.

GSEA analysis of discrete subsets of genes exclusively regulated by Six1 or Eya1 suggested that they also regulate some categories of genes independently of one another. A particularly interesting finding was the extreme enrichment of *Hox* genes (specifically of the Antennapedia-type) in the Eya1-specific subset of up-regulated genes (*Figure 3F*), suggesting that Eya1 may play a previously un-identified role in regulating Hox gene expression independently of Six1.

## Characterisation of transcriptional regulators activated by Six1 or Eya1

To verify our RNA-Seq data, we selected a number of target genes for further characterisation and, in order to gain insight into the GRN downstream of Six1 and Eya1, we restricted candidates to transcription factors or co-factors up-regulated by Six1 or Eya1. First, we generated a list of well-supported target genes containing all genes with at least a two-fold up-regulation in at least two of our three treatment groups (*Table 1*). From the 228 genes in this list we selected all 30 transcription factors or co-factors for further analysis. However, we were unable to amplify two genes from this list (*Egr3, Fbxo41*) from cDNA and therefore omitted these genes from further characterisation. We additionally included *Sox3* and *Ngn1* - which were found to be slightly below our threshold of two-fold up-regulation in at least two treatment groups - because previous studies have implicated these genes in the regulation of placodal neurogenesis downstream of Six1 and Eya1 (*Ma et al., 1996*, *1998*; *Schlosser et al., 2008*; *Ahmed et al., 2012b*) (*Table 2*).

The expression of genes previously undescribed in *Xenopus* (*Crem, FosB, Hes8, Isl2, Znf214*) was fully characterised in neural fold, and early and late tail bud stages, along with those for which expression has been described for relatively few stages (*Atoh1, Emx1.2, Gfi1a, Hes2, Hes9, Lhx5, Mab21l2b, Pou3f2b, Pou4f1.2, Ripply3, Sim1, Sox21, Tbx6, Tlx1*) (summarised in *Figure 4A–T*; *Figure 4—figure supplements 1–5*). Genes with extensively characterised expression patterns

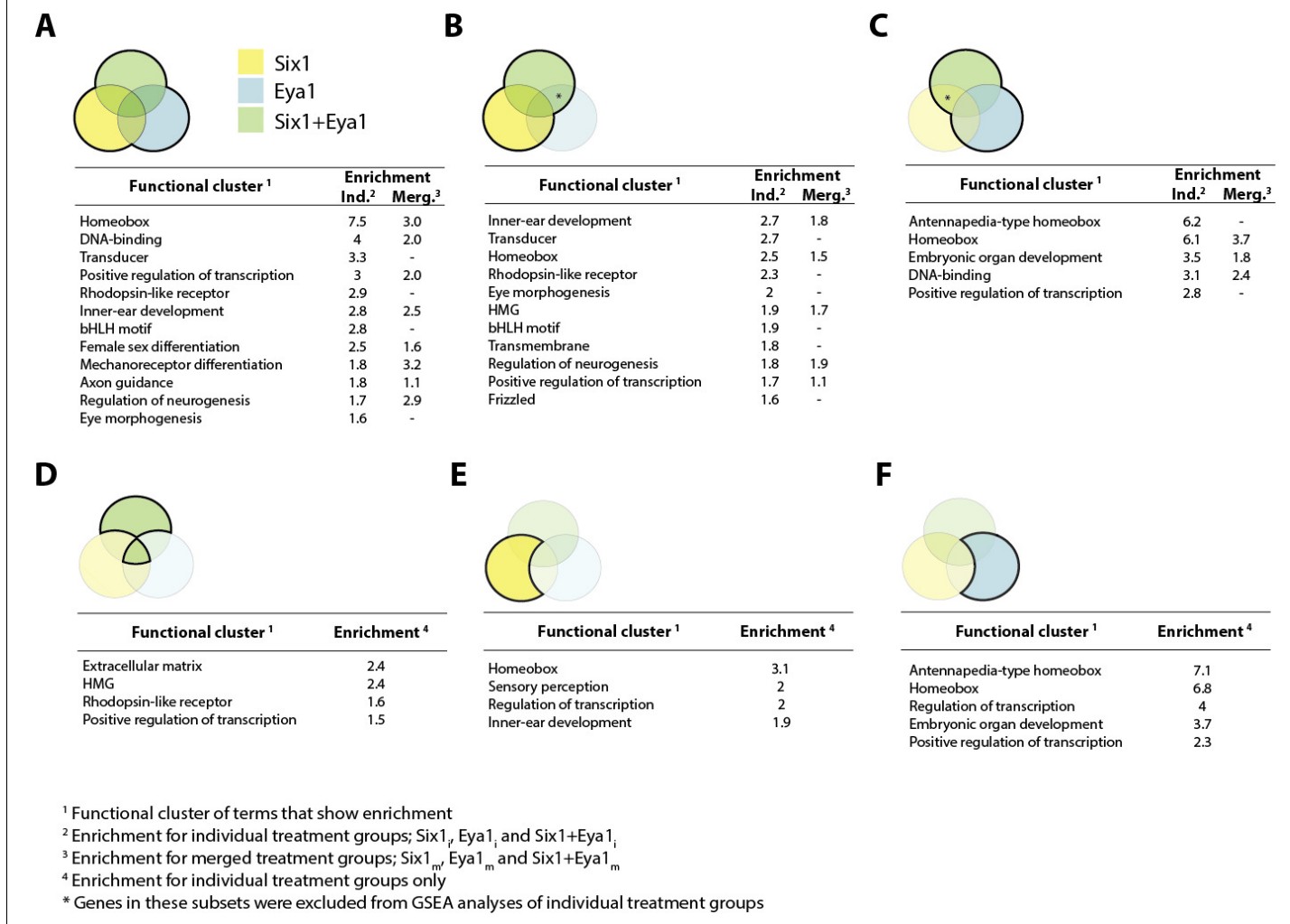

**Figure 3.** Gene set enrichment analysis (GSEA) showing enriched clusters of functional terms for up-regulated genes in different treatment groups. In each case, treatment groups considered are highlighted and outlined in bold in the accompanying Venn diagram. Yellow colouring indicates Six1 treatment; blue shows Eya1 and green Six1+Eya1. Enrichment scores ≥1.5 are reported for individual treatment groups (Ind.) and, where available, ≥0.5 for merged treatment groups (Merg.). (A) Up-regulated genes from all treatment groups included in analysis; (B) *Six1* overexpression only; (C) *Eya1* overexpression only. (D) Genes differentially expressed after overexpression of both *Six1* and *Eya1*; (E) exclusively after *Six1* overexpression; (F) exclusively after *Eya1* overexpression.

The following figure supplement is available for figure 3:

**Figure supplement 1.** Gene set enrichment analysis (GSEA) showing enriched clusters of functional terms for down-regulated genes in different treatment groups.

(*Ngn1* (*Nieber et al., 2009*); *Six1* (*Pandur and Moody, 2000*); *Six2* (*Ghanbari et al., 2001*); *Sox2* (*Mizuseki et al., 1998*); *Sox3* (*Penzel et al., 1997*); *Sox17* (*Hudson et al., 1997*), *MyoD1* (*Hopwood et al., 1989*), *Sox1* (*Nitta et al., 2006*), *Ets2a* (*Salanga et al., 2010*), *Mafa* (*Coolen et al., 2005*)) are not shown here.

We found 19/30 (63.3% ) of these transcriptional regulators to be expressed in PPE or placodal derivatives, while 11/30 (*Sox17, MyoD1, Sox1, Ets2a, Mafa, Emx1.2, Lhx5, Pou3f2b, Tbx6, Tbx15, Sim1)* were not expressed in either the PPE or any placodal derivatives. However, many of the genes in the latter group were expressed in the adjacent neural folds or other tissues. Thus, it is possible that such genes may be direct targets of Six1 or Eya1 in domains surrounding the PPE, likely to have

**Table 1.** Genes with at least two-fold up-regulation in at least two out of three individual treatment groups (Six1$_i$; Eya1$_i$; Six1+Eya1$_i$).

| | Annotation* | Accession | Six1 FC[†] | Eya1 FC[‡] | Six1+Eya1 FC[§] |
|---|---|---|---|---|---|
| | Chromosome unknown open reading frame | XM_002938866.2 | 6.2 | - | 7.9 |
| | cDNA clone IMAGE:7022272 | BC094950.1 | 5.6 | 5.1 | 7.5 |
| | *X. laevis* cyclin Dx (ccndx) | NP_001087887.1 | 7.5 | - | 5.2 |
| | Calcitonin gene-related peptide-like | XM_002941675.2 | 7 | - | 3 |
| | *X. laevis* tripartite motif containing 63, E3 ubiquitin protein ligase (trim63) | NM_001093214.1 | 5.3 | 3.6 | 6.3 |
| | ATP-sensitive inward rectifier potassium channel 11-like | XM_004916278.1 | 5.1 | - | 6.1 |
| | Leucine rich repeat containing 52 (lrrc52) | XM_002933773.2 | 6.1 | - | 2.8 |
| # | SIX homeobox 2 (six2) | NM_001100275.1 | 5 | 3.5 | 5.9 |
| | Potassium voltage-gated channel shaker-related subfamily member 2 (kcna2) | XM_004910736.1 | 5.1 | - | 4.9 |
| | Butyrophilin subfamily 2 member A1 (btn2a1) | NM_001094508.1 | - | 1.2 | 4.9 |
| | Glutathione peroxidase 2 (gpx2) | NM_001256315.1 | - | 2.5 | 4.8 |
| # | *X. laevis* for Xsox17-alpha protein | AJ001730.1 | 3.6 | 2.6 | 4.8 |
| | *X. laevis* ectodysplasin A receptor (edar) | NM_001087047.1 | 2.8 | 2.5 | 4.7 |
| | Uncharacterized (LOC101734405) | XM_004918247.1 | 4.4 | 0.8 | 3.5 |
| | Glutathione peroxidase 2 (gpx2) | NM_001256315.1 | 3.4 | 2.1 | 4.4 |
| | *X. laevis* cytochrome P450, family 2, subfamily D, polypeptide 6 (cyp2d6) | NM_001093574.1 | 1.1 | - | 4.4 |
| | Calcium/calmodulin-dependent protein kinase kinase 2beta (camkk2) | XM_002937701.2 | 4.4 | 2.6 | 3.5 |
| | Cytochrome P450 family 26 subfamily B polypeptide 1 (cyp26b1) | NM_001079187.2 | 3.3 | 4 | 4.3 |
| | Troponin I type 1 (skeletal, slow) | BC061268 | 1.8 | - | 4.3 |
| | 72 kDa inositol polyphosphate 5-phosphatase-like (LOC101734556) | XM_004916572.1 | - | 4.2 | 1.3 |
| | Chemokine (C-X-C motif) receptor 7 (cxcr7) | NM_001030434.1 | 3 | 2.8 | 4.1 |
| # | *X. laevis* xSox17 alpha 2 | AB052691.1 | 1.7 | 1.4 | 4 |
| | Espin (espn) transcript variant X3 | XM_004916193.1 | - | 2.1 | 4 |
| | B-cell CLL/lymphoma 11B (zinc finger protein) (bcl11b) | XM_004917116.1 | - | 1.9 | 4 |
| | C-X-C motif chemokine 10-like | XM_002940578.2 | 1.9 | 4 | 3.5 |
| | *X. laevis* hedgehog-interacting protein | BC046952.1 | - | 2.7 | 4 |
| | X-linked inhibitor of apoptosis (xiap) | NM_001030412.1 | 4 | 3.1 | 2.3 |
| | *X. laevis* uncharacterized (LOC496300) | NM_001095458.1 | 1.4 | 3.9 | 1.1 |
| | *X. laevis* RDC1 like protein | BC098974.1 | 3.6 | 2.1 | 3.9 |
| | *X. laevis* for frizzled 4 protein (fz4 gene) | AJ251750.1 | 1.3 | 0.6 | 3.8 |
| | Espin (espn) transcript variant X1 | XM_002933856.2 | 3.1 | 2.7 | 3.7 |
| | Paired box 1 (pax1) transcript variant X1 | JQ929179.1 | - | 3 | 3.7 |
| | Potassium voltage-gated channel subfamily F member 1 (kcnf1) | NM_001102926.1 | 3.6 | - | 2.1 |
| | Echinoderm microtubule-associated protein-like 1-like | XM_004917169.1 | - | 3.6 | 2.8 |
| | Leucine rich adaptor protein 1-like (lurap1l) | XM_002940127.2 | 3.6 | 1.4 | 2.4 |
| | Sine oculis binding protein homolog (Drosophila) | BC154687.1 | 2.7 | 1.6 | 3.4 |
| | RNA-directed DNA polymerase homolog | XM_004916122.1 | 3.4 | - | 2.1 |
| | Kinesin family member 3C (kif3c) transcript variant X1 | XM_004914940.1 | 1.4 | 0.8 | 3.4 |
| | Anoctamin 2 (ano2) | XM_002932297.2 | 2.2 | 1.3 | 3.4 |
| | *X. laevis* natriuretic peptide C (nppc) | NM_001112924.1 | 2.1 | - | 3.3 |
| | Uncharacterized (LOC101734952) | XM_004916172.1 | 2.5 | - | 3.3 |
| | Poly (ADP-ribose) polymerase 14-like (LOC101731378) | XM_004920062.1 | 3.1 | - | 3.3 |
| | Protocadherin-11 X-linked-like (LOC100493938) | XM_004916890.1 | - | 3.2 | 1.4 |
| | Uncharacterized (LOC101733225) | XM_004919937.1 | 2.5 | 3.2 | 1.6 |

*Table 1 continued on next page*

*Table 1 continued*

| | Annotation* | Accession | Six1 FC† | Eya1 FC‡ | Six1+Eya1 FC§ |
|---|---|---|---|---|---|
| | Calcium channel voltage-dependent beta 4 subunit (cacnb4) | NM_001142151.1 | 3.1 | - | 1.9 |
| | F-box protein 32 (fbxo32) transcript variant X1 | XM_002941397.2 | 1.8 | - | 3.1 |
| | cDNA clone TEgg026p17 | CR761997.2 | 3 | 2.6 | - |
| | *X. laevis* transforming growth factor beta-induced (tgfbi) | NM_001095238.1 | 1.3 | - | 3 |
| | Mucin-2-like (LOC100494747) | XM_002936043.2 | 3 | 2 | 1.7 |
| | *X. laevis* uncharacterized protein (MGC68450) | NM_001089841.1 | 2.2 | - | 2.8 |
| | *X. laevis* neuregulin alpha-1 | AF076618.1 | 1.4 | 0.8 | 2.7 |
| | Potassium voltage-gated channel Isk-related (kcne1) | XM_004912135.1 | 2.2 | 1.5 | 2.7 |
| | Olfactory receptor 5G3-like (LOC100492086) | XM_002942220.1 | 1.9 | - | 2.7 |
| | Alpha-kinase 2 (alpk2) | XM_004910401.1 | 1.1 | 2.2 | 2.7 |
| | *X. laevis* arginyl amino peptidase (amino peptidase B) b (rnpep-b) | NM_001092079.1 | - | 2.7 | 1.7 |
| # | *X. laevis* SRY-box containing protein (Sox1) | EF672727.1 | - | 2.6 | 2.1 |
| | Copine II (cpne2) transcript variant X1 | XM_004913481.1 | 1 | 1.2 | 2.6 |
| | *X. laevis* hemoglobin, gamma A (hbg1) | NM_001096347 | 1.2 | - | 2.6 |
| | KIAA0895 protein (kiaa0895) | NM_001114073.1 | 1.6 | 2.6 | - |
| # | *X. laevis* empty spiracles homeobox 1gene 2 (emx1.2) | NM_001093430.1 | 2.6 | 1.9 | 1.1 |
| | Homeobox B8 (hoxb8) transcript variant X1 | XM_002938021.2 | 1.1 | 2.5 | - |
| | Monocyte to macrophage differentiation-associated (mmd) | XM_004918560.1 | - | 1.2 | 2.5 |
| | *X. laevis* uncharacterized (LOC100036933) | NM_001097704.1 | 1.5 | 1.5 | 2.5 |
| | Finished cDNA clone TNeu143f19 | CR760056.2 | 2.2 | 2.5 | - |
| | Chromosome unknown open reading frame C2orf80 | XM_002937119.2 | 1.4 | 2.1 | 2.4 |
| # | Single-minded homolog 1 (sim1) transcript variant X2 | XM_004914545.1 | - | 1.4 | 2.4 |
| | Transmembrane protein 2-like (LOC100491930) | XM_002932255.2 | 2.4 | 1.9 | 1.3 |
| | PX domain containing 1 (pxdc1) | NM_001130262.1 | 1.4 | - | 2.4 |
| | Aldehyde dehydrogenase 1 family member L2 (aldh1l2) | XM_002938070.2 | 0.9 | 1.3 | 2.3 |
| | Uncharacterized (LOC100490228) | XM_002942932.2 | 1.8 | - | 2.3 |
| | Beta-1 3-galactosyltransferase 2-like (LOC101732799) | XM_004918863.1 | 1.6 | 2.3 | - |
| | Alpha-2 3-sialyltransferase ST3Gal V (st3gal5) | FN550108.1 | 1.8 | - | 2.3 |
| | *X. laevis* uncharacterized protein (MGC64538) | NM_001086337.1 | - | 1.6 | 2.3 |
| | Transmembrane channel-like protein 7-like (LOC100493700) | XM_002932222.2 | 1.4 | 0.9 | 2.3 |
| | Kinase insert domain receptor (a type III receptor tyrosine kinase) (kdr) | XM_002934669.2 | 1.9 | 0.9 | 2.3 |
| | Serine/threonine kinase 32A (stk32a) | XM_002936707.2 | 1.3 | 2.2 | - |
| | Pancreatic lipase-related protein 2 (pnliprp2) | NM_001089647.1 | 2.1 | 0.7 | 2.2 |
| | *X. laevis* nephrin (NPHS1) | AY902238.1 | - | 2.2 | 1.1 |
| | Poly (ADP-ribose) polymerase 14-like (LOC100485144) | XM_002943546.2 | 2 | 2.2 | 1.2 |
| | Frizzled family receptor 4 (fzd4) | XM_002936543.2 | 1.4 | 0.7 | 2.1 |
| | Neuropeptide Y receptor Y2 (npy2r) | XM_004911153.1 | 2.1 | - | 1.6 |
| | Deoxyribonuclease gamma-like (LOC100497175) | XM_002938386.2 | 1.8 | 2.1 | 2 |
| | *X. laevis* dehydrogenase/reductase (SDR family) member 11 (dhrs11) | NM_001094963.1 | - | 1.5 | 2.1 |
| | *X. laevis* gamma-glutamyl hydrolase (ggh) | NM_001092691.1 | 2.1 | 1.3 | 2 |
| | Opsin-3-like | XM_002932623.2 | 1 | 1.2 | 2 |
| | *X. laevis* transmembrane protein 56 (tmem56-b) | NM_001086447.1 | - | 1.1 | 2 |
| | *X. laevis* pyruvate dehydrogenase phosphatase catalytic subunit 1 (pdp1) | NM_001094221.1 | 1.5 | 2 | 1 |
| | ArfGAP with SH3 domain ankyrin repeat and PH domain 3 (asap3) | XM_002939360.2 | 1.7 | - | 1.9 |

*Table 1 continued on next page*

*Table 1 continued*

| | Annotation* | Accession | Six1 FC† | Eya1 FC‡ | Six1+Eya1 FC§ |
|---|---|---|---|---|---|
| # | Early growth response 3 (egr3) | XM_002932703.2 | 1.6 | 0.8 | 1.9 |
| # | POU class 4 homeobox 1 (pou4f1.2) | NM_001097307.1 | 1.3 | 1 | 1.9 |
| | Activin beta B subunit | S61773.1 | - | 1.7 | 1.8 |
| | Monocyte to macrophage differentiation-associated (mmd) | XM_002937811.2 | 1.7 | 1.1 | 1.8 |
| | X. laevis ribosomal protein S2e | BC130122.1 | - | 1.8 | 1.7 |
| | X. laevis ras homolog family member V (rhov) | NM_001128659.1 | 1.2 | 0.8 | 1.6 |
| | X. laevis adenomatosis polyposis coli down-regulated 1 (apcdd1) | NM_001094109.1 | 1.2 | 1 | 1.6 |
| # | X. laevis zinc finger protein 214 (znf214) | NM_001097042.1 | 1.2 | 0.8 | 1.5 |
| | X. laevis cdc25Ba | AB363840.1 | 1.2 | - | 1.5 |
| | X. laevis apelin (apln-a) | NM_001097924.1 | 0.9 | 1.3 | 1.5 |
| | Suppressor of cytokine signaling 2 (socs2) | NM_001095760.1 | - | 1.1 | 1.5 |
| # | cAMP responsive element modulator (crem) | XM_002935162.2 | - | 1.4 | 1.5 |
| | X. laevis clone IMAGE:4684003 | BC042305.1 | 1.4 | - | 1.2 |
| # | X. laevis ets-2a proto-oncogene | BC133183.1 | 1.3 | 1 | 1.4 |
| | X. laevis similar to envoplakin | BC045116.1 | 1.4 | 1.4 | - |
| | Ras homolog family member V (rhov) | NM_001095566.1 | 1.4 | 1 | 1.2 |
| | Samd9l protein (samd9l) | XM_002943522.2 | - | 1.2 | 1.3 |
| | Flocculation protein FLO11-like (LOC100490389) | XM_002942555.2 | 1.2 | - | 1.3 |
| | c-Jun-amino-terminal kinase-interacting protein 4-like (LOC100493724) | XM_002939963.2 | 1.1 | - | 1.2 |
| | X. laevis Dickkopf-1 (Xdkk-1) | AF030434.1 | 1 | 1.2 | 1.1 |
| | X. laevis ectoderm neural cortex related-3 (Encr-3) | AY216793.1 | 1.1 | 0.8 | 1.2 |
| | Uncharacterized (LOC101730819) | XM_004915204.1 | 0.9 | 1.1 | 1.2 |
| # | X. laevis LIM class homeodomain protein | BC084744.1 | 1.1 | 0.7 | 1.1 |
| | Ceramide kinase-like (cerkl) | XM_002932015.2 | 1.4 | 1.3 | 2 |
| | Mannosyl (alpha-1 3-)-glycoprotein beta-1 4-N-acetylglucosaminyltransferase (mgat4b) | NM_001091975.1 | 2 | - | 1.8 |
| | Fibroblast growth factor 19 (fgf19) | NM_001142825.1 | - | 2 | 1.5 |
| # | F-box protein 41 (fbxo41) | NM_001079043.1 | 1.3 | 0.6 | 2 |
| | Avidin-like (LOC100487365) | XM_002939983.2 | 2 | 1.6 | - |
| | Four and a half LIM domains 2 (fhl2) | NM_001126761.1 | - | 1.1 | 1.9 |
| | Metalloprotease TIKI1-like (LOC100491951) | XM_002936336.2 | 1.1 | 1.4 | 1.9 |
| | X. laevis Kazal-type serine peptidase inhibitor domain 1 (kazald1) | NM_001092073.1 | 1.6 | 1.1 | 1.9 |
| | Uncharacterized (LOC101734664) | XM_004910525.1 | 1.2 | 0.6 | 1.9 |
| | X. laevis similar to calsequestrin 2 (cardiac muscle) | BC097545.1 | 1.8 | 1.5 | 1.9 |
| | X. laevis COMM domain containing 3 (commd3) | NM_001095386.1 | 1.1 | 1.9 | 0.6 |
| | X. laevis alcohol dehydrogenase iron containing1 (adhfe1) | NM_001127802.1 | - | 1.9 | 1.2 |
| | X. laevis ectonucleoside triphosphate diphosphohydrolase 1 (entpd1) | NM_001092268.1 | 1.8 | 0.6 | 1.3 |
| # | Protein fosB-like transcript variant X2 | XM_004916957.1 | - | 1.7 | 1.4 |
| | Tocopherol (alpha) transfer protein (ttpa) | NM_001008184.1 | - | 1.7 | 1.6 |
| | X. laevis tetratricopeptide repeat domain 39B (ttc39b) | NM_001094701.1 | 1.1 | - | 1.7 |
| # | X. laevis Tbx6 (Tbx6) | DQ355794.1 | 1.4 | 1.7 | 1 |
| | X. laevis uncharacterized (LOC100036989) | NM_001097746.1 | - | 1.3 | 1.7 |
| | X. laevis cDNA clone IMAGE:6947552 | BC093552.1 | 1.3 | 1.7 | - |
| | B-cell CLL/lymphoma 10 (bcl10) | NM_001015777.2 | 1.7 | - | 1.2 |
| | Uncharacterized (LOC100494710) | XM_002939048.2 | 1.4 | 1.6 | - |

*Table 1 continued on next page*

Table 1 continued

| | Annotation* | Accession | Six1 FC[†] | Eya1 FC[‡] | Six1+Eya1 FC[§] |
|---|---|---|---|---|---|
| | *X. laevis* keratin 17 (krt17) | NM_001094941.1 | - | 1.2 | 1.6 |
| | Membrane metallo-endopeptidase-like 1 (mmel1) | NM_001127095.1 | 0.9 | 1.1 | 1.6 |
| | Putative methyltransferase KIAA1456 homolog | XM_002934674.2 | 1.1 | - | 1.6 |
| | Phospholipase Cdelta 3 (plcd3) | XM_002935518.2 | 1.1 | 1.5 | 1.6 |
| | IdnK gluconokinase homolog (E. coli) (idnk) | NM_001126592.1 | 1.4 | 0.9 | 1.5 |
| | Uncharacterized (LOC100486093) transcript variant X2 | XM_002939117.2 | 1.5 | - | 1.5 |
| | *X. laevis* similar to calsequestrin 2 (cardiac muscle) | BC041283.1 | 1.1 | - | 1.5 |
| | Piwi-like RNA-mediated gene silencing 2 (piwil2) | NM_001112999.1 | 1.1 | - | 1.5 |
| | Zinc finger and BTB domain containing 20 (zbtb20) | XM_002939649.2 | 1.4 | - | 1.1 |
| # | V-maf musculoaponeurotic fibrosarcoma oncogene homolog A (mafa) | NM_001032304.1 | 1.4 | 0.9 | 1.1 |
| | *X. laevis* uncharacterized protein (MGC81120) | NM_001091225.1 | 1.4 | 0.9 | 1.3 |
| # | Single-minded homolog 1 (*Drosophila*) (sim1) transcript variant X3 | XM_004914546.1 | 1.1 | 1.3 | 1.2 |
| | Xenopus laevis alpha-2-macroglobulin-like 1 (a2ml1) | NM_001135077.1 | 1.1 | - | 1.1 |
| | *X. laevis* chromogranin A (parathyroid secretory protein 1) (chga) | NM_001094724.1 | 1.6 | 1.4 | 2.2 |
| | *X. laevis* lipaseendothelial (lipg) | NM_001090061.1 | 1.2 | 1.3 | 0.6 |
| | G protein-coupled receptor 56 (gpr56) | XM_002931653.2 | 1.7 | - | 1.6 |
| | *X. laevis* family with sequence similarity 101member B (fam101b) | NM_001093870.1 | 1.5 | 0.8 | 1.5 |
| | *X. laevis* CD81 protein (cd81-a) | NM_001086613.1 | 0.7 | 1.1 | 1.9 |
| | *X. laevis* calbindin D28k | BC170542.1 | 2.2 | - | 3.1 |
| | *X. laevis* ATPaseNa+/K+ transportingbeta 1 polypeptide (atp1b1) | NM_001086759.1 | 1.2 | 1 | 1.7 |
| | *X. laevis* 7-transmembrane receptor frizzled-1 | AF231711.1 | 1.4 | 1 | 2 |
| | *X. laevis* prostaglandin reductase 2 (ptgr2) | NM_001079334.1 | 1.4 | 1.5 | - |
| | *X. laevis* TGF-beta2 for transforming growth factor-beta2 | X51817.1 | 1.3 | - | 1.1 |
| # | SRY (sex determining region Y)-box 2 (sox2) | NM_213704.3 | 1.1 | 1.3 | 1.9 |
| # | *X. laevis* for enhancer of split related 9 (esr9 gene) | AJ009282.1 | 1.7 | 1.6 | - |
| | *X. laevis* mal T-cell differentiation protein (mal) | NM_001086577.1 | - | 1.2 | 1.4 |
| | Transmembrane proteaseserine 13 (tmprss13) | XM_002932904.2 | 1.5 | 1.1 | 1.9 |
| | *X. laevis* Ras-related associated with diabetes (rrad) | NM_001092750.1 | 8 | 4.6 | 4.2 |
| | Integrin beta 4 (itgb4) transcript variant X1 | XM_002939974.2 | 1.4 | - | 2.2 |
| | *Xenopus (Silurana) tropicalis* FERM domain containing 4A (frmd4a) | XM_002935243.2 | 1.1 | 0.6 | 1.3 |
| | *X. laevis* complement factor I (cfi-a) | NM_001085952.1 | 1.4 | 1.2 | 1.6 |
| # | *X. laevis* SIX homeobox 1 (six1) | NP_001082027.1 | 1.4 | 1.2 | 2.3 |
| | FH2 domain-containing protein 1-like (LOC100496216) | XM_002934907.2 | 1.9 | 0.9 | 1.9 |
| # | *X. laevis* mab-21-like 2 (mab21l2-b) | NM_001096770.1 | - | 2.8 | 2.9 |
| | *X. laevis* regulator of cell cycle (rgcc) | NM_001093976.1 | 1.3 | 1.1 | 1.7 |
| | *X. laevis* Cep63 | FJ464988.1 | - | 1.4 | 2.3 |
| | *X. laevis* CD81 antigen (target of anti proliferative antibody 1) | BC041217.1 | 1.7 | 1.1 | 2 |
| | Transmembrane serine protease 9 | BC087611.1 | 1.1 | 1.1 | 1.2 |
| # | *X. laevis* POU class 3 homeobox 2 (pou3f2-b) | NM_001096751.1 | 3 | 2.3 | 2.9 |
| | G protein-coupled receptor 153 (gpr153) | NM_001128052.1 | 2.5 | 1.1 | 1.5 |
| # | *X. laevis* Myoblast determination protein 1 homolog A | BC041190.1 | 3.5 | 2.7 | 4.7 |
| # | T-cell leukemia homeobox 1 (tlx1) transcript variant 1 | XM_002936768.2 | 2.6 | 2.3 | 2.6 |
| | *X. laevis* neurotrophin 3 (ntf3) | NM_001092740.1 | 1.4 | 1.5 | 1.9 |
| | *X. laevis* p21 GTPase-associated kinase 1 (PAK1) | AF000239.1 | 1.2 | - | 2.1 |

Table 1 continued on next page

*Table 1 continued*

| | Annotation* | Accession | Six1 FC† | Eya1 FC‡ | Six1+Eya1 FC§ |
|---|---|---|---|---|---|
| # | *X. laevis* hairy and enhancer of split 9, gene 1 (hes9.1-b) | NP_001089097.1 | 1.8 | 1.5 | 1.6 |
| | *X. laevis* tetraspanin 1 (tspan1) | NM_001095473.1 | 1.2 | 0.7 | 1.3 |
| | *X. laevis* uncharacterized protein (MGC83079) | NM_001091250.1 | 2 | 1.5 | - |
| | *X. laevis* cDNA clone IMAGE:5085355 | BC073731.1 | 1.3 | - | 1.4 |
| | Family with sequence similarity 198member A (fam198a) | XM_002937853.2 | 1.7 | 0.7 | 1.3 |
| | Progestin and adipoQ receptor family member IX (paqr9) | XM_004914351.1 | 1.7 | - | 1.2 |
| # | Hairy and enhancer of split 8 (hes8) | XM_002933849.2 | 2.8 | 1.7 | 3.6 |
| | *X. laevis* p21 GTPase-associated kinase 1 | BC081113.1 | 1.3 | 0.8 | 1.7 |
| | Finished cDNA clone TNeu008g03 | CR761907.2 | 1.2 | 1.1 | 0.7 |
| | WD repeat domain 27 (wdr27) | XM_002931515.2 | 1.2 | 2.2 | 1.1 |
| # | Growth factor independent 1 transcription repressor (gfi1) | XM_002933803.2 | 1.8 | 1.8 | 3.2 |
| | Protein phosphatase 2 regulatory subunit B'beta (ppp2r5b) | NM_001100279.1 | 2.4 | 1.4 | 4.2 |
| | Ornithine decarboxylase antizyme 2 (oaz2), transcript variant 2 | NP_001106583.2 | 1.8 | - | 1.5 |
| | *X. laevis* fast troponin T (TNNT3) | AY114144.1 | - | 1.1 | 1.5 |
| # | *X. laevis* xRipply3 for xRipply3 protein | AB455086.1 | 0.9 | 1.1 | 2 |
| | RAS-like family 11member B (rasl11b) | NM_001015774.1 | - | 1.2 | 1.4 |
| | *X. laevis* for thimet oligopeptidase | BC070748.1 | 3.8 | - | 2 |
| | *X. laevis* fibroblast growth factor 3 (fgf3) | NM_001008153.1 | 2 | 1.2 | 2 |
| | *X. laevis* cDNA clone IMAGE:8332229 | BC155363.1 | 1.5 | 0.9 | 1.4 |
| | Proline rich 15 (prr15) | XM_002933381.2 | 1.6 | - | 1.3 |
| | Integrin beta 6 (itgb6) | NM_001097306.1 | 2.3 | 0.6 | 2.8 |
| # | *Xenopus laevis* empty spiracles homeobox 1, gene 2 (emx1.2) | NM_001093430.1 | 2.6 | 1.4 | 2.1 |
| | *X. laevis* p21-activated kinase (PAK1) | AF169794.1 | 1.4 | 1.8 | 2.6 |
| # | ISL LIM homeobox 2 (isl2) | NM_001166041.1 | 1.5 | - | 1.7 |
| # | Atonal homolog 1 (Drosophila) (atoh1) | XM_004911085.1 | 0.9 | 1.1 | 1.5 |
| | Ectodysplasin A receptor (edar) | NM_001087047.1 | 4.3 | - | 3.3 |
| | *X. laevis* degr03 | DQ096846.1 | 2.1 | 2.2 | 2 |
| | Calcyphosine (caps) | NM_001097320.1 | - | 1.4 | 3.7 |
| | *X. laevis* kiaa0930 | NM_001086221.1 | 1.5 | 1 | 1.6 |
| | Putative N-acetyltransferase 16-like (LOC100490742) | XM_002943189.1 | 2.1 | 1 | 1.7 |
| # | T-box 15 (tbx15) | XM_002940981.2 | 2 | 1 | 1.8 |
| # | SRY (sex determining region Y)-box 1 (sox1) | NM_001080996.1 | 0.6 | 1.5 | 1.2 |
| | Cytochrome P450 family 2 subfamily C polypeptide 18 (cyp2c18) | NM_001091776.1 | 2.1 | 1.4 | 1.6 |
| | *X. laevis* calcitonin receptor-like (calcrl) | NM_001086737.1 | 1.1 | 0.8 | 1.6 |
| | *X. laevis* claudin 3 (cldn3) | NM_001005709.1 | 2.1 | 1.3 | 1.5 |
| | Atlastin GTPase 1 (atl1) | NM_001078754.1 | 1.8 | 2 | 1.7 |
| | Rho GTPase activating protein 9 (arhgap9), transcript variant X2 | XM_012957829 | 1.8 | 1.2 | 3.4 |
| # | *X. laevis* Hes2 | BC084134.1 | 1.7 | 0.9 | 1.3 |
| | *X. laevis* U3 snRNA | X07318.1 | 1 | 2.8 | 1.1 |
| | Uncharacterized (LOC101732195) | XM_004912378.1 | 2 | - | 1.5 |
| | Tumor necrosis factor receptor superfamilymember 21 (tnfrsf21) | NM_001079136.1 | 1.1 | 0.8 | 1.2 |
| | *X. laevis* arginase 3 | U08408.1 | - | 1.3 | 1.8 |
| | ChaC cation transport regulator homolog 1 (chac1) | XM_002939546.2 | 1.2 | 1.3 | 1.5 |
| | *X. laevis* DIRAS familyGTP-binding RAS-like 3 (diras3) | NM_001095243.1 | 0.8 | 1.7 | 1.4 |
| | *X. laevis* DnaJ (Hsp40) homolog subfamily C member 27 (dnajc27-b) | NM_001095422.1 | 1.1 | 0.8 | 1.1 |

* Genes are ranked by FC value, using the highest FC in each of the three treatment groups. Genes included must have FC ≥ 1 in at least two out of the three treatment groups as well as showing at least a two-fold difference in FC to the un-injected control (not shown). Corresponding values ≥0.5 are shown for all treatments.

† Log$_2$ Fold change values after Six1-GR overexpression.

‡ Log$_2$ Fold change values after Eya1-GR overexpression.

§ Log$_2$ Fold change values after Six1-GR+Eya1-GR overexpression.

# Denotes transcription factors with at least a two-fold change in at least two treatment groups selected for further analysis.

been included in our screen as a result of non-PPE contamination during dissection. Of the transcriptional regulators identified in our list of well-supported targets and expressed in the PPE or placodal derivatives 79% (15/19) were statistically supported in the analyses of merged datasets (*Table 2*). These included genes broadly expressed in cranial ectoderm including the PPE (*Crem, FosB, Znf214, Ripply3*), and genes expressed in the PPE or subdomains of the PPE and subsequently in some placodes (*Hes2, Hes8, Hes9, Mab21l2b, Six1, Six2, Sox2, Sox3, Sox21, Atoh1, Ngn1, Gfi1a, Isl2, Pou4f1.2, Tlx1*) (*Figure 4A–T*).

To begin to elucidate the GRN downstream of Six1 and Eya1 we chose ten transcription factors showing expression in posterior placodes (i.e. those derived from the posterior placodal area; the lateral line, otic and epibranchial placodes) for additional functional studies including genes implicated in the maintenance of neuronal progenitors (*Sox2, Sox3, Hes8* and *Hes9*) as well as genes implicated in the regulation of sensory or neuronal differentiation (*Atoh1, Gfi1a, Isl2, Ngn1, Pou4f1.2* and *Tlx1*). Selected genes were independently verified as being direct targets of either Six1 (*Isl2*) or of both Six1 and Eya1 (all other targets; *Sox3* not analysed) in the PPE by qPCR, and the results were broadly consistent with the RNA-Seq data (*Figure 4U and V*).

## Six1 and Eya1 are required for expression of transcriptional regulators of neurogenesis in the PPE and placodes

To explore whether Six1 or Eya1 were required for the expression of selected target genes, the expression of each target was analysed by in-situ-hybridisation after MO-mediated knockdown of *Six1* or *Eya1*. The efficacy and specificity of both co-injected *Six1*-MOs (*Six1*-MO1 and *Six1*-MO2; *Brugmann et al., 2004*) and *Eya1*-MOs (*Eya1*-MO1 and *Eya1*-MO2; *Schlosser et al., 2008*) have been previously reported. Compared to injection with a control MO (*Eya1*-mmMO with 5 mismatches relative to Eya1-MO2), knockdown of either *Six1* or *Eya1* significantly reduced the expression of all direct target genes in PPE or placodes, demonstrating that both Six1 and Eya1 are required for their expression (*Figure 5* and *Figure 5—figure supplement 1*; *Table 3*). To control for off-target effects associated with MO use, target gene expression was also analysed after overexpression of a dominant-negative version of *Six1* (*Six1-EnR*; *Brugmann et al., 2004*). Expression patterns of all target genes were highly similar to those seen after MO-knockdown of either *Six1* or *Eya1*, suggesting that the observed reductions in expression were caused by *Six1* or *Eya1* knockdown as opposed to being an artefact of MO use (*Figure 5—figure supplement 2*). Taken together, these findings show that Six1 and Eya1 are essential direct upstream regulators of multiple genes encoding transcription factors that promote neuro- and sensorigenesis in the PPE and placodes.

## Six1 and Eya1 affect expression of presumptive direct target genes in complex ways

To complement the loss-of-function studies, and to examine the spatial distribution of presumptive direct targets of Six1 and Eya1 in gain-of-function experiments, we injected Six1-GR and Eya1-GR individually and, to ensure that overexpression did not affect early embryogenesis, induced their nuclear translocation by adding DEX at neural fold stage (stages 16–18), after PPE commitment (*Ahrens and Schlosser, 2005*). Surprisingly, although injection of Six1-GR or Eya1-GR resulted in up-regulation of direct targets in a minority of cases (*Table 3*; *Figures 6* and *7*), the dominant observed phenotype was down-regulation of target gene expression in the PPE or placodes (*Table 3*; *Figures 8* and *9*). Considering that here, unlike in the initial RNA-Seq screen and qPCR experiments, CHX was not used to block protein synthesis, these results indicate that Six1 and Eya1 additionally affect expression of many of their direct target genes in indirect and partly opposing ways.

**Table 2.** Transcription factors and co-factors selected for characterisation by in-situ-hybridisation ranked by FC value in individual treatment.

| Annotation | Gene | Accession | Individual | | | Merged | | |
|---|---|---|---|---|---|---|---|---|
| | | | Six1* | Eya1[†] | Six1 +Eya1[‡] | Six1[§] | Eya1[#] | Six1 +Eya1[¶] |
| SIX homeobox 2 (Six2) | Six2 | NM_001100275.1 | 5 | 3.5 | 5.9 | 5.4** | 5** | 4.9** |
| X. laevis for Xsox17-alpha protein | Sox17 | AJ001730.1 | 3.6 | 2.6 | 4.8 | 4.4** | 3.3** | 3.5** |
| X. laevis Myoblast determination protein 1 homolog A | MyoD1 | BC041190.1 | 3.5 | 2.7 | 4.7 | 4.1** | 4.2** | 3.9** |
| Hairy and enhancer of split 8 (Hes8) | Hes8 | XM_002933849.2 | 2.8 | 1.7 | 3.6 | 3.2** | 3.2** | 3.1** |
| Growth factor independent 1 transcription repressor (Gfi1) | Gfi1a | XM_002933803.2 | 1.8 | 1.8 | 3.2 | 2.4** | 2.6** | 4.1** |
| X. laevis POU class 3 homeobox 2 (Pou3f2-b) | Pou3f2b | NM_001096751.1 | 3 | 2.3 | 2.9 | 3** | 2.6** | 2.7** |
| X. laevis Mab-21-like 2 (Mab21l2-b) | Mab21l2b | NM_001096770.1 | - | 2.8 | 2.9 | - | - | - |
| T-cell leukemia homeobox 1 (Tlx1) transcript variant 1 | Tlx1 | XM_002936768.2 | 2.6 | 2.3 | 2.6 | 2.4** | 2.4** | 2.4** |
| X. laevis empty spiracles homeobox 1 gene 2 (Emx1.2) | Emx1.2 | NM_001093430.1 | 2.6 | 1.9 | 1.1 | - | - | 1.7** |
| X. laevis SRY-box containing protein (Sox1) | Sox1 | EF672727.1 | - | 2.6 | 2.1 | - | 2** | - |
| Single-minded homolog 1 (Sim1) transcript variant X2 | Sim1 | XM_004914545.1 | - | 1.4 | 2.4 | - | - | - |
| X. laevis SIX homeobox 1 (Six1) | Six1 | AF279254.1 | 1.4 | 1.2 | 2.3 | 1.9** | 1.6** | 1.6** |
| F-box protein 41 (Fbxo41) | Fbxo41 | NM_001079043.1 | 1.3 | 0.6 | 2 | - | - | - |
| T-box 15 (Tbx15) | Tbx15 | XM_002940981.2 | 2 | 1 | 1.8 | 2** | 1.4** | 1.7** |
| X. laevis xRipply3 for xRipply3 protein | Ripply3 | AB455086.1 | 0.9 | 1.1 | 2 | 1.6** | 1.4** | 1.3** |
| Early growth response 3 (Egr3) | Egr3 | XM_002932703.2 | 1.6 | 0.8 | 1.9 | 1.7** | 1.3** | 1.9** |
| SRY (sex determining region Y)-box 2 (Sox2) | Sox2 | NM_213704.3 | 1.1 | 1.3 | 1.9 | 1.6** | 1.6** | 1.5** |
| POU class 4 homeobox 1 (Pou4f1.2) | Pou4f1.2 | NM_001097307.1 | 1.3 | 1 | 1.9 | 1.6** | 1.5** | 1.5** |
| X. laevis for enhancer of split related 9 (Esr9 gene) | Hes9.1a | AJ009282.1 | 1.7 | 1.6 | - | - | - | - |
| ISL LIM homeobox 2 (Isl2) | Isl2 | NM_001166041.1 | 1.5 | - | 1.7 | 1.6** | 1.1** | 1.4** |
| X. laevis Tbx6 (Tbx6) | Tbx6 | DQ355794.1 | 1.4 | 1.7 | 1 | - | - | - |
| Protein FosB-like transcript variant X2 | FosB | XM_004916957.1 | - | 1.7 | 1.4 | - | 1.4** | 1.2** |
| X. laevis Hes2 | Hes2 | BC084134.1 | 1.7 | 0.9 | 1.3 | - | - | - |
| cAMP responsive element modulator (Crem) | Crem | XM_002935162.2 | - | 1.4 | 1.5 | - | 1.4** | 1.2** |
| X. laevis zinc finger protein 214 (Znf214) | Znf214 | NM_001097042.1 | 1.2 | 0.8 | 1.5 | 1.2** | 5.9** | 5.8** |
| Xenopus laevis SRY (sex determining region Y)-box 21 (Sox21) | Sox21 | NM_001172213.1 | 1.2 | 0.6 | 1.5 | 1.4** | 1.2** | 1.2** |
| Atonal homolog 1 (Drosophila) (Atoh1) | Atoh1 | XM_004911085.1 | 0.9 | 1.1 | 1.5 | 1 | 1 | 1 |
| X. laevis Ets-2a proto-oncogene | Ets2a | BC133183.1 | 1.3 | 1 | 1.4 | 1.3** | 1.2** | 1.2** |
| V-maf musculoaponeurotic fibrosarcoma oncogene homolog A (Mafa) | Mafa | NM_001032304.1 | 1.4 | 0.9 | 1.1 | 1.9** | - | 1.8** |
| X. laevis LIM class homeodomain protein | Lhx5 | BC084744.1 | 1.1 | - | 1.1 | - | - | - |
| Xenopus (Silurana) tropicalis neurogenin 1 (Neurog1) | Ngn1 | NM_001123423.1 | 0.8 | 0.9 | 0.8 | 0.8 | 0.8 | 0.8** |
| Xenopus laevis SOX3 protein | Sox3 | BC072222.1 | 0.5 | - | 0.9 | 0.7 | 0.7 | 0.6 |

*Log$_2$ fold change values after *Six1* overexpression (Six1$_i$).

[†] Log$_2$ fold change values after *Eya1* overexpression (Eya1$_i$).

[‡] Log$_2$ fold change values after *Six1+Eya1* overexpression (Six1+ Eya1$_i$).

[§] Log$_2$ fold change values after overexpression of *Six1* or *Six1+Eya1* (Six1$_m$).

[#] Log$_2$ fold change values after overexpression of *Eya1* or *Six1+Eya1* (Eya1$_m$).

[¶] Log$_2$ fold change values after overexpression of *Six1* or *Eya1* or *Six1+Eya1* (Six1+Eya1$_m$).

** Denotes statistically supported data ($q < 0.05$).

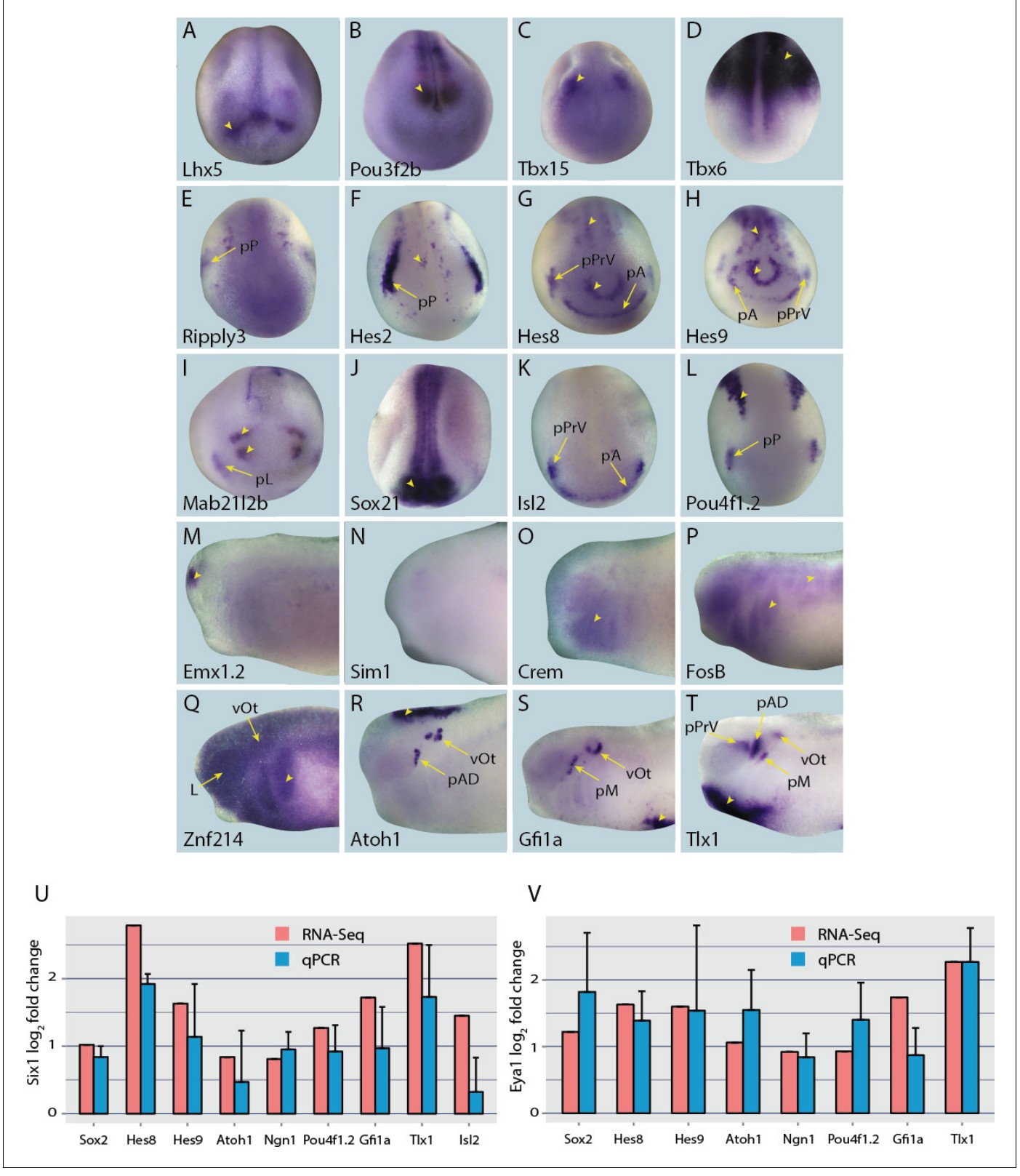

**Figure 4.** Expression of selected presumptive direct targets of Six1/Eya1 in whole-mount *Xenopus* embryos. Genes expressed at neural plate stages (stages 14–18) are shown in panels **A–L**, and those only expressed at later stages are shown at mid/late tail bud stage (stages 28–32) in panels **M–T**. Several of the genes surveyed (*Lhx5, Pou3f2b, Tbx15, Tbx6, Emx1.2* and *Sim1* (**A–D**, **M** and **N**), were not expressed in the PPE, nor any placodal derivatives in later stages. Instead, such genes were expressed in the adjacent neural folds (*Lhx5, Pou3f2b and Tbx15*), ectoderm (*Tbx6),* or in the

*Figure 4 continued on next page*

*Figure 4 continued*

forebrain at later stages (*Emx1.2*). Several other genes were expressed broadly across the cranial ectoderm, at least partially overlapping with the PPE at neural plate stages (*Ripply3, Crem, FosB* and *Znf214*; **E,O–Q**), some of which are also maintained in placodal derivatives such as *Znf214* in the otic vesicle (**Q**). The remaining genes (**F–L, R–T**) are expressed in parts of the PPE and maintained in some placodes (*Hes2, Hes8, Hes9, Mab21l2b, Sox21, Isl2, Pou4f1.2,* and *Tlx1*) or are expressed in a subset of placodes only (*Atoh1, Gfi1a*) (see *Figure 4—figure supplements 1–4* for additional stages). Yellow arrows mark placodal expression. Arrowheads mark non-placodal expression. Abbreviations: pA: anterior placodal region; pAD: anterior lateral line placode; pE: epibranchial placode; pL: lens placode; L: lens; pM: middle lateral line placode; pO: olfactory placode; pP: posterior placodal region; pPrV: profundal/trigeminal placodes; vOt: otic vesicle. Plots **U** and **V** show qPCR after *Six1* or *Eya1* overexpression. Log$_2$ fold change values were calculated from qPCR data obtained after overexpression of Six1-GR (**U**) or Eya1-GR (**V**) in placodal explants and are shown next to corresponding fold change values obtained from the RNA-Seq data. In all cases shown, qPCR values broadly corroborate those from the RNA-Seq data - showing up-regulation of target genes after either *Six1* or *Eya1* overexpression. Vertical error bars show the standard deviation of the mean of biological triplicates.

The following figure supplements are available for figure 4:

**Figure supplement 1.** Expression of targets not expressed in placodes in whole-mount *Xenopus* embryos.

**Figure supplement 2.** Expression of targets that broadly overlap with PPE in whole-mount *Xenopus* embryos.

**Figure supplement 3.** Expression of targets with dynamic/transient expression pattern in placodes in whole-mount *Xenopus* embryos.

**Figure supplement 4.** Expression of targets with persistent expression in placodes in whole-mount *Xenopus* embryos.

**Figure supplement 5.** Expression of selected targets in pre-placodal ectoderm (PPE) in sections through neural plate stage *Xenopus* embryos.

## Discussion

Overexpression of GR-fusion constructs followed by DEX-induced nuclear translocation in the presence of protein synthesis inhibitors has been previously used successfully to screen for direct target genes of transcription factors or cofactors in *Xenopus* (*Kolm and Sive, 1995*; *Taverner et al., 2005*; *Seo et al., 2007*). Here, we combine this approach with high-throughput sequencing of tissue-specific RNA to identify several hundred novel presumptive direct target genes of Six1 and Eya1 in the PPE. We show that this strategy indeed recovers the majority of direct Six1 target genes known from previous studies, indicating its reliability. Our in situ and qPCR analyses of target genes predicted from the RNA-Seq screen also provided independent verification of selected target genes suggesting a low false discovery rate. Moreover, the expression of all genes selected for detailed analysis proved to be dependent on Six1 and Eya1 in the PPE, indicating that many genes our screen predicted as Six1/Eya1 targets are also functionally dependent on these upstream regulators in the PPE. A comparison of our data set with recently identified direct target genes of *sine oculis*, the *Six1* orthologue in the developing eye of *Drosophila* (*Jusiak et al., 2014*; *Jin et al., 2016*) also reveals that homologues to six out of the 12 *sine oculis* target genes identified with high confidence in *Jin et al. (2016)* are differentially expressed in our Six$_i$ or Six1+Eya1$_i$ (and often also in Eya1$_i$) treatment groups, viz. *Six1* and *Six2; Eya4; Shh*; various matrix metalloproteases (e.g. *MMP9*); *Ets2*; and *Frizzled1* and *Frizzled4*. This suggests that a relatively high proportion of Six1 target genes may be evolutionarily conserved.

Our finding that many of the presumptive direct target genes of Six1 or Eya1 are not up-regulated in the absence of CHX indicates that without blocking protein synthesis it is not possible to reliably identify direct target genes, presumably due to the existence of indirect interactions with such targets. We believe that this is one of the reasons why our findings differ substantially from the study of *Yan et al. (2015)*, which analysed differentially expressed genes in *Xenopus* animal cap explants after overexpression of Six1 without first blocking protein synthesis. None of the transcription factors in our prioritised list was identified in the study by *Yan et al. (2015)*; and we found none of the transcription factors differentially expressed in their study. A second likely reason for the discrepancy between the results presented here and in *Yan et al. (2015)* is that, while we specifically analysed PPE tissue (presumably containing tissue-specific cofactors required for the activation or repression of Six1 and Eya1 target genes specific for the developing placodes), they analysed target genes in animal cap tissue, known to be composed of pluripotent cells.

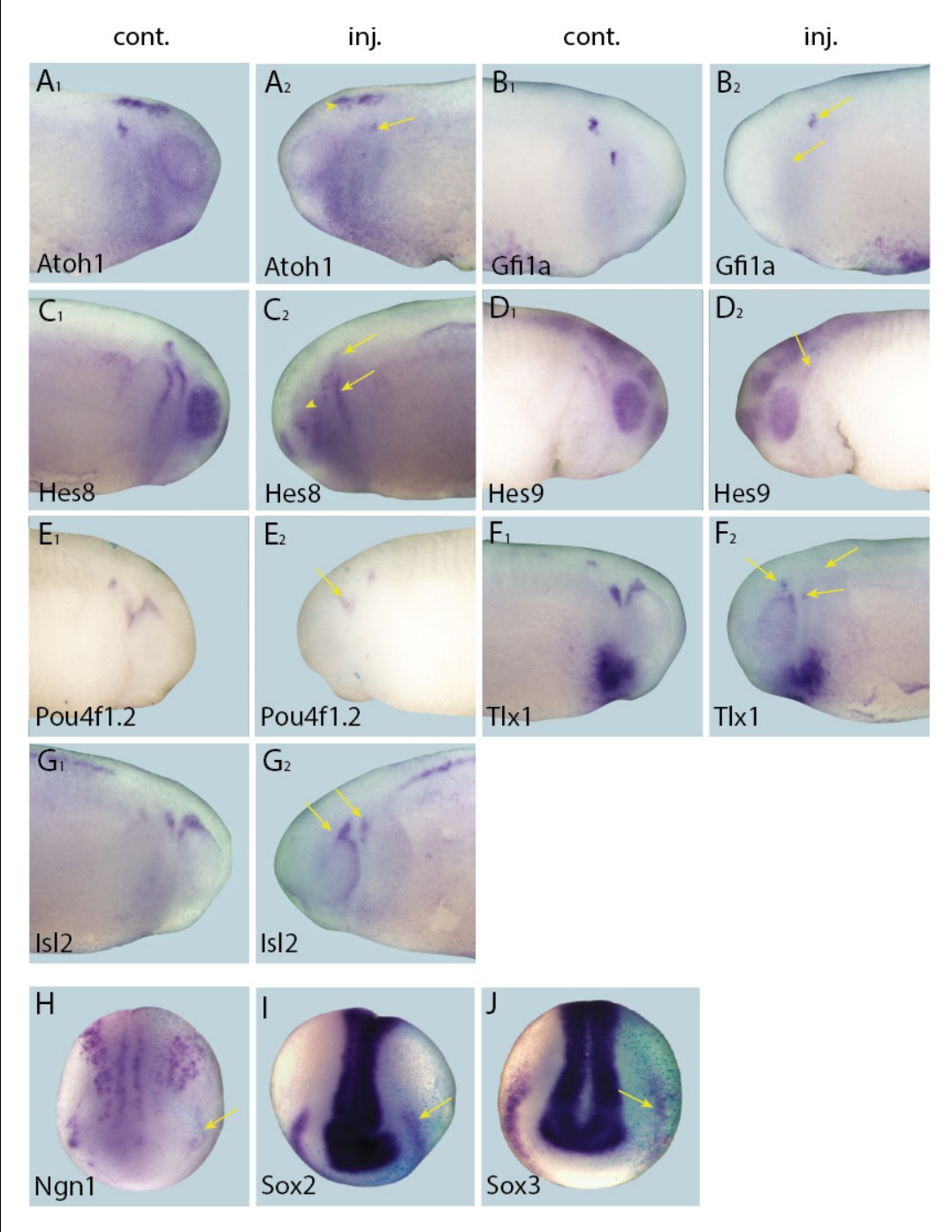

**Figure 5.** Effects of *Eya1* knockdown on target genes. Tail bud (**A–G**) and neural plate (**H–I**) stage embryos after unilateral injection of *Eya1*-MO1+2. In each case, *lacZ* was co-injected as a lineage tracer and panels **A₁–G₁** show the control (un-injected) side and **A₂–G₂** show the injected side (*lacZ* staining out of frame in most specimens). The injected side is positioned to the right in **H–J**, as marked by blue *lacZ* staining. Arrows and arrowheads mark reductions in marker gene expression in placodal and non-placodal derivatives, respectively.

The following figure supplements are available for figure 5:

**Figure supplement 1.** Effects of Six1 knockdown on target genes.

**Figure supplement 2.** Repression of Six1 target genes by *Six1-EnR* injection.

**Table 3.** Changes in marker gene expression in the placodes after injection of various constructs.

| | Injection | *Six1*-MO* | *Eya1*-MO* | Six1-EnR | *Eya1*-mmMO | Six1-GR§ | Eya1-GR§ |
|---|---|---|---|---|---|---|---|
| | | % | % | % | % | % | % |
| | Phenotype | (n) | (n) | (n) | (n) | (n) | (n) |
| Atoh1 | Reduced | 77** | 90‡ | 94 | 10 | 26 | 42 |
| | | (26) | (20) | (18) | (21) | (19) | (12) |
| | Increased | 0 | 0 | 0 | 0 | 35 | 42 |
| | | (26) | (20) | (18) | (21) | (17) | (12) |
| Gfi1a | Reduced | 82† | 67† | 69 | 31 | 57 | 36 |
| | | (27) | (17) | (16) | (26) | (14) | (14) |
| | Increased | 0 | 0 | 0 | 0 | 7 | 43 |
| | | (27) | (17) | (16) | (26) | (14) | (14) |
| Hes8 | Reduced | 74‡ | 83‡ | 70 | 17 | 60 | 57 |
| | | (35) | (35) | (46) | (24) | (40) | (56) |
| | Increased | 0 | 0 | 24 | 0 | 15 | 29 |
| | | (35) | (35) | (46) | (24) | (40) | (56) |
| Hes9 | Reduced | 73‡ | 76‡ | 84 | 11 | 75 | 29 |
| | | (45) | (33) | (38) | (27) | (12) | (29) |
| | Increased | 0 | 0 | 8 | 0 | 0 | 0 |
| | | (45) | (33) | (38) | (27) | (12) | (29) |
| Isl2 | Reduced | 66† | 100‡ | nd | 27 | 50 | 24 |
| | | (38) | (17) | nd | (22) | (18) | (17) |
| | Increased | 6 | 0 | nd | 0 | 31 | 41 |
| | | (38) | (17) | nd | (22) | (16) | (17) |
| Ngn1 | Reduced | 65‡ | 49† | 84 | 17 | 17 | 36 |
| | | (51) | (43) | (31) | (24) | (30) | (59) |
| | Increased | 0 | 16 | 6 | 4 | 23 | 41 |
| | | (51) | (43) | (31) | (24) | (30) | (59) |
| Pou4f1.2 | Reduced | 67‡ | 63† | 71 | 16 | 47 | 81 |
| | | (48) | (30) | (35) | (19) | (15) | (37) |
| | Increased | 0 | 0 | 0 | 0 | 13 | 0 |
| | | (48) | (30) | (35) | (19) | (15) | (37) |
| Sox2 | Reduced | 74‡ | 78‡ | 87 | 6 | 90 | 48 |
| | | (19) | (18) | (30) | (16) | (21) | (33) |
| | Increased | 0 | 0 | 23# | 0 | 0 | 12 |
| | | (19) | (18) | (30) | (16) | (21) | (33) |
| Sox3 | Reduced | 68‡ | 54† | 39 | 9 | 49 | 40 |
| | | (25) | (26) | (31) | (22) | (25) | (23) |
| | Increased | 0 | 0 | 71# | 0 | 16 | 17 |
| | | (25) | (26) | (31) | (22) | (25) | (23) |
| Tlx1 | Reduced | 84† | 91‡ | 100 | 33 | 40 | 7 |
| | | (31) | (32) | (13) | (15) | (10) | (15) |
| | Increased | 6 | 0 | 0 | 0 | 40 | 73 |
| | | (31) | (32) | (13) | (15) | (10) | (15) |

* Significant differences (Fisher's exact test);

† p<0.05,

‡ p<0.001) to *Eya1*-mmMO injections are indicated.

§ Dexamethasone treatment from stages 16–18 on.
# Expression ectopic in epidermis.
n: Number of embryos analysed at both neural plate (stage 14–16) and tail bud (stage 21–26) stage.
nd: Not determined.

Previous studies have shown that Six1 and Eya1 are essential for both the establishment of the PPE (*Brugmann et al., 2004*; *Christophorou et al., 2009*), as well as for the subsequent development of placode-derived sense organs (*Xu et al., 1999*; *Laclef et al., 2003*; *Zheng et al., 2003*; *Brugmann et al., 2004*; *Zou et al., 2004*; *Kozlowski et al., 2005*; *Schlosser et al., 2008*; *Ahmed et al., 2012b*, *2012a*) but the mechanisms through which they act are poorly understood. The continued expression of both genes in almost all placodes developing from the PPE (*Schlosser and Ahrens, 2004*), combined with the observed deficiencies in derivatives from most placodes after loss-of-function of either Six1 or Eya1, indicates that they play a role in generic aspects of placode development shared by all placodes. Indeed, our data show that genes revealed as presumptive direct targets of Six1 and Eya1 were highly enriched for GO terms associated with neurogenesis and placode development. Our screen also confirms previous studies suggesting that Six1 and Eya1 synergistically regulate many genes in the PPE, and that the Six1-Eya1 protein complex predominantly acts by activating transcription (*Li et al., 2003*; *Brugmann et al., 2004*). However, we also find support for independent action of Six1 and Eya1 in the PPE, possibly in

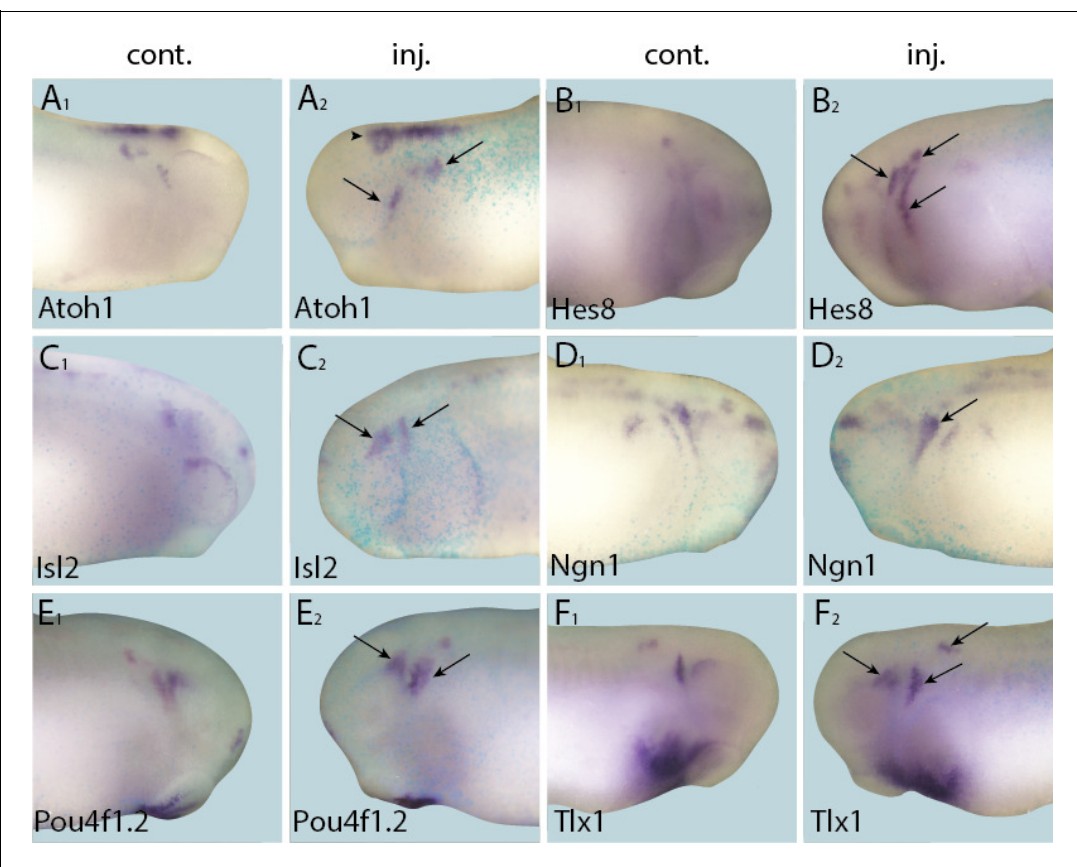

**Figure 6.** Up-regulation of target gene expression domains after overexpression of Six1. Tail bud stage embryos (**A–F**) after unilateral injection of *Six1*-GR and DEX induction at neural plate stage (16–18). In each case, *lacZ* was co-injected as a lineage tracer and panels **A₁–F₁** show the control (un-injected) side and **A₂–F₂** show the injected side. Arrows and arrowheads mark expansions in marker gene expression in placodal and non-placodal derivatives, respectively.

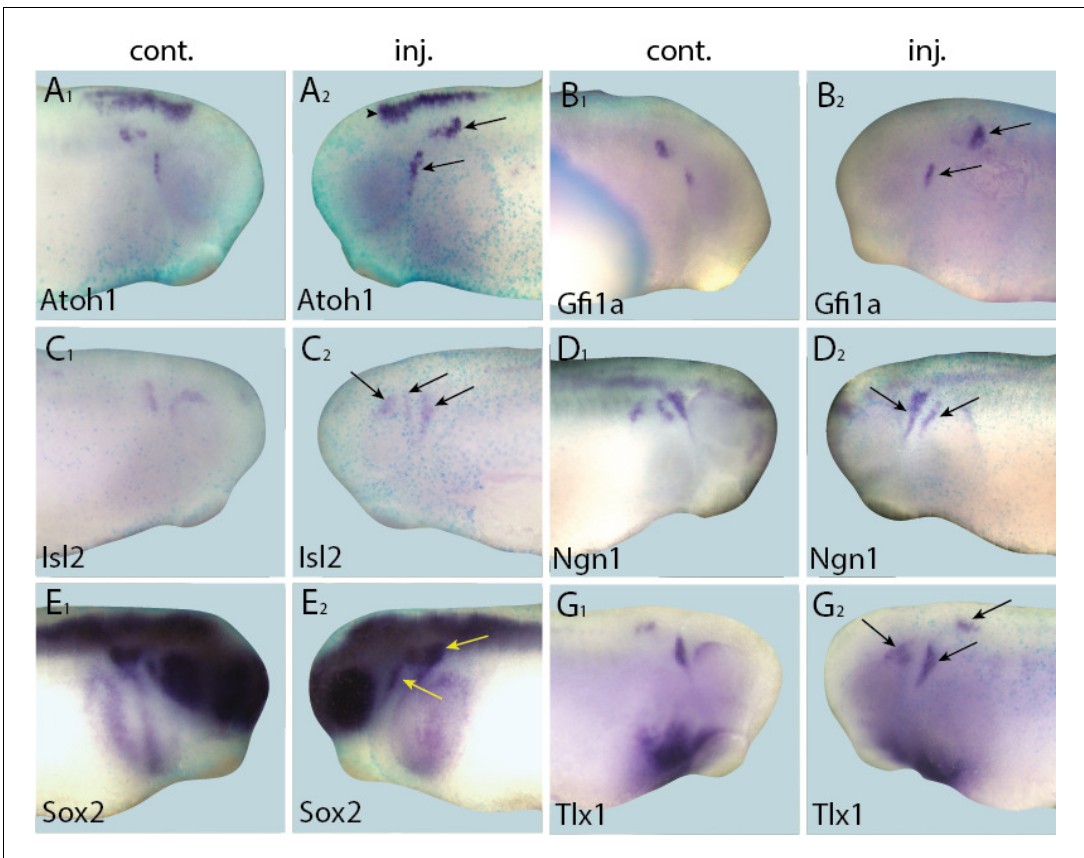

**Figure 7.** Up-regulation of target gene expression domains after overexpression of Eya1. Tail bud stage embryos (A–G) after unilateral injection of *Eya1*-GR and DEX induction at neural plate stage (16–18). In each case, *lacZ* was co-injected as a lineage tracer and panels $A_1$–$G_1$ show the control (un-injected) side and $A_2$–$G_2$ show the injected side. Arrows and arrowheads mark expansions in marker gene expression in placodal and non-placodal derivatives, respectively.

conjunction with other interacting partners (*Brugmann et al., 2004*; *Ahmed et al., 2012a*). Surprisingly, we found *Hox* genes to be strongly enriched in the list of target genes activated by Eya1 only. This deserves further study since Eya1 has not previously been recognised as an upstream regulator of *Hox* genes.

It has previously been suggested (*Schlosser, 2006*) that a generic role of Six1 and Eya1 for all placodes could be implemented in two possible ways: (1) By the direct contribution to the activation of genes regulating developmental processes shared between different placodes such as proliferation, morphogenetic movements and neuronal or sensory cytodifferentiation; or (2) by direct contribution to the activation of genes defining the identity of different individual placodes within the PPE. Our data strongly suggest that Six1 and Eya1 act predominantly in the first rather than in the second mode. A large number of transcription factor encoding genes, including several *Pax*, *Pitx*, *ANF* and *FoxI* genes, have been implicated in conferring identity to individual placodes, or groups of placodes, within the PPE (reviewed in *Schlosser, 2006*, *2010*; *Grocott et al., 2012*; *Saint-Jeannet and Moody, 2014*) however only a few of these genes were recovered as targets of Six1 or Eya1, e.g. *Gbx2* (FC 1.7 in Six1+Eya1$_i$) and *FoxI4* (FC 1.09 in Six1$_i$). In contrast, we found a large number of genes encoding transcription factors with roles in neuronal/sensory cytodifferentiation but also other proteins with likely roles for the maintenance of proliferating progenitors (e.g. *Cyclin D*, *RGCC*), the regulation of cell adhesion and morphogenetic movements (e.g. *EDAR*, *CXCR7*, *Protocadherin11*, *RhoV*, *Claudin3*) and cytodifferentiation (e.g. *Espin*, *Neurotrophin3*). This suggests that, similar to *Hox* or *Pax* genes, *Six1* and *Eya1* act as both master genes and micro-managers

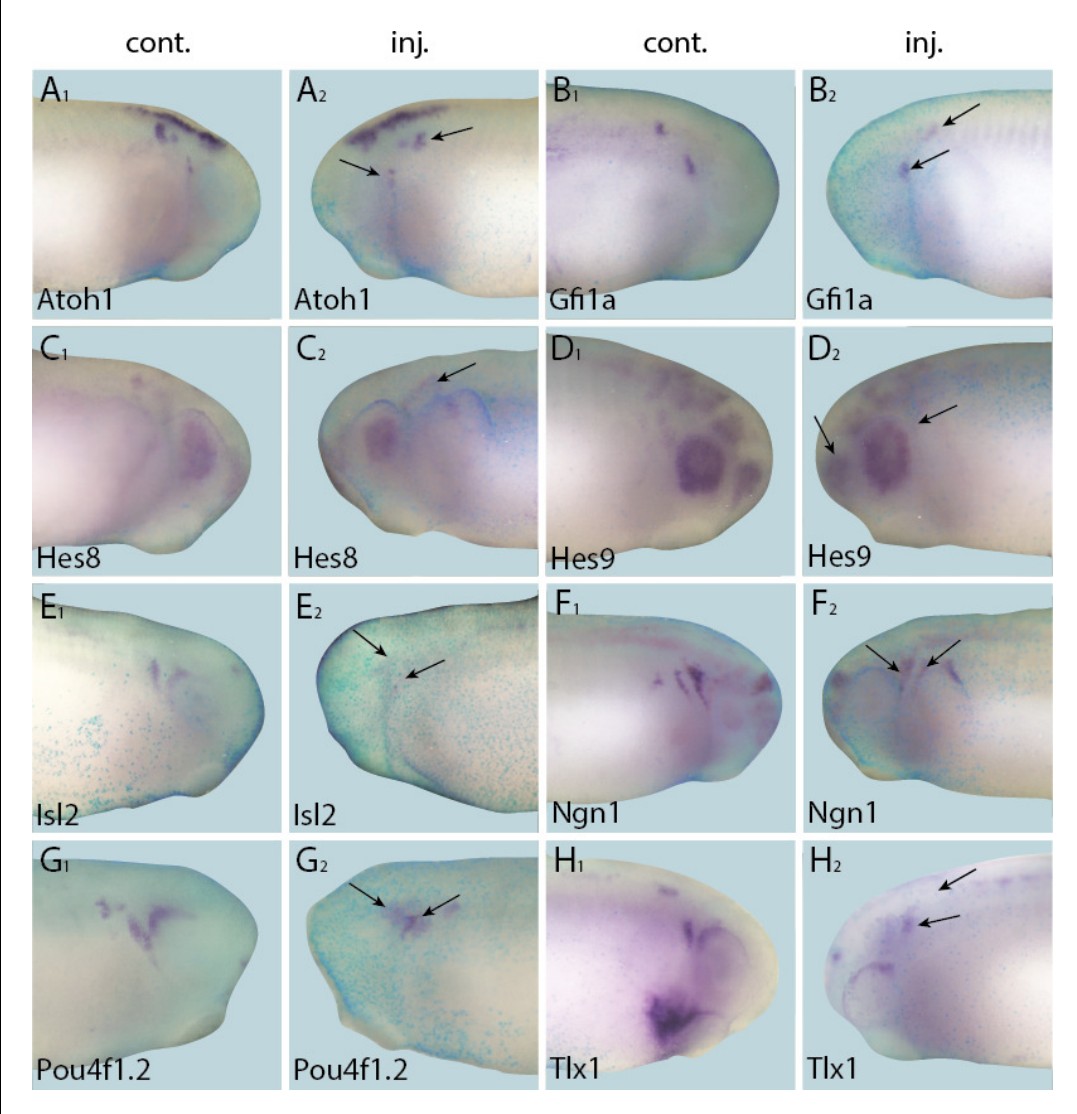

**Figure 8.** Down-regulation of target gene expression domains after overexpression of Six1. Tail bud stage embryos (A–H) after unilateral injection of *Six1*-GR and DEX induction at neural plate stage (16–18). In each case, *lacZ* was co-injected as a lineage tracer and panels $A_1$–$H_1$ show the control (un-injected) side and $A_2$–$H_2$ show the injected side. Arrows mark reductions in marker gene expression in placodal derivatives.

(*Akam, 1998*; *Thompson and Ziman, 2011*; *Rezsohazy et al., 2015*), acting upstream of a GRN co-ordinating cell differentiation in the PPE as well as directly activating terminal differentiation gene batteries.

Considering that Six1 and Eya1 have previously been shown to promote a proliferative progenitor state at high doses but neuronal and sensory differentiation at lower doses (*Schlosser et al., 2008*), it is particularly interesting that we identified presumptive direct target genes encoding transcription factors previously implicated in progenitor maintenance (Sox2, Sox3, Hes8, Hes9) and differentiation (Ngn1, Atoh1, POU4f1, Gfi1a, Isl2, Tlx1). Both Hes (Hes8, Hes9) and SoxB1 (Sox2, Sox3) proteins are known to keep progenitor cells in an undifferentiated state, and must be down-regulated for neuronal differentiation to proceed. While Sox2 and Sox3 play multiple roles including activity as pioneer factors, which open up chromatin for transcription (*Bylund et al., 2003*; *Graham et al., 2003*; *Pevny and Placzek, 2005*; *Bergsland et al., 2011*), Hes proteins generally repress neuronal/sensory determination genes such as *Ngn1* or *Atoh1* as effectors of Notch signalling (*Kobayashi and*

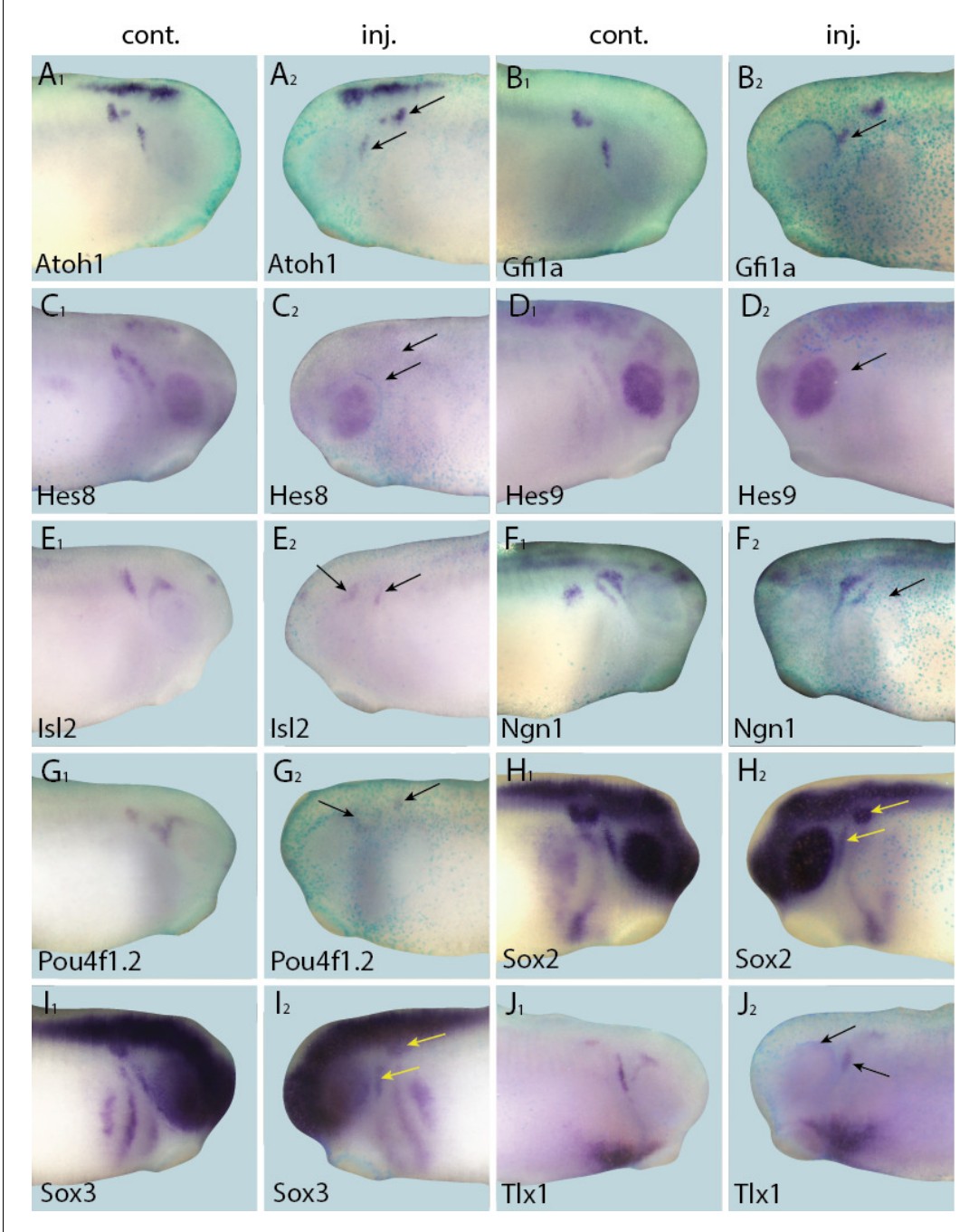

**Figure 9.** Down-regulation of target gene expression domains after overexpression of Eya1. Tail bud stage embryos (**A–J**) after unilateral injection of *Eya1*-GR and DEX induction at neural plate stage (16–18). In each case, *lacZ* was co-injected as a lineage tracer and panels **A₁–J₁** show the control (un-injected) side and **A₂–J₂** show the injected side. Arrows mark reductions in marker gene expression in placodal derivatives.

*Kageyama, 2014*; *Su et al., 2015*; *Abdolazimi et al., 2016*). Conversely, Ngn1 and Atoh1 are known to act as proneural factors that initiate differentiation of sensory neurons and hair cells, respectively (*Ma et al., 1996*, *1998*; *Bermingham et al., 1999*; *Millimaki et al., 2007*), whereas POU4f1 (previously known as Brn3a), Gfi1a, Isl2 and Tlx1 act further downstream in differentiation of sensory neurons (*Patterson and Krieg, 1999*; *Wallis et al., 2003*; *Cheng et al., 2004*; *Eng et al.,*

*2004*; *Lanier et al., 2009*; *Dykes et al., 2011*), and Gfi1a and the related POU domain factor POU4f3 (or Brn3c) are required for hair cell maintenance and survival (*Xiang et al., 1998*; *Wallis et al., 2003*). Our findings strongly indicate that Six1 and Eya1 directly promote multiple steps during sensory and neuronal development, and act to drive both progenitor maintenance and neuronal differentiation programmes in placodes (summarised in *Figure 10*), although further functional studies are needed to clarify the mechanism allowing Six1 and Eya1 to maintain the balance

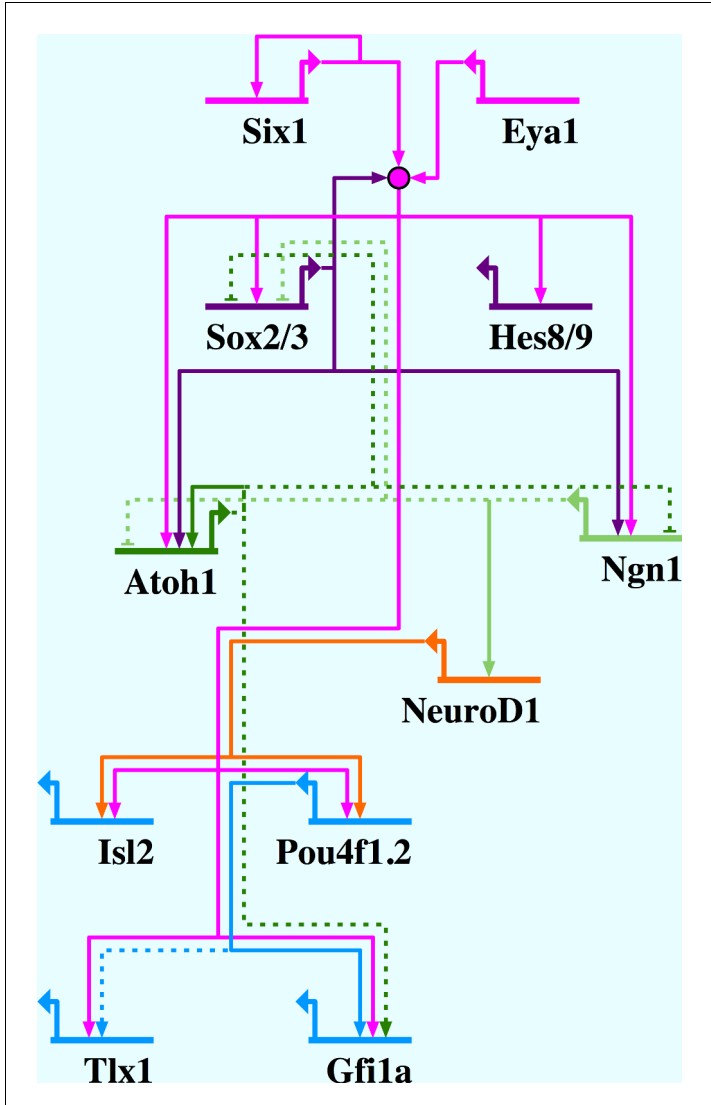

**Figure 10.** Network summary for Six1/Eya1-activated gene regulation in the PPE. Six1/Eya1 act to promote neuronal differentiation, by activation of pro-neural genes (*Ngn1, Atoh1*), as well as progenitor state maintenance, by activation of genes such as *SoxB1* and *Hes* genes. Arrows indicate direct (solid line) and indirect (dotted line) activation; barred lines show direct (solid line) and indirect (dotted line) repression. Evidence for interactions: Six1 positively autoregulates (*Sato et al., 2012*); Six1/Eya1 directly activate Sox2, Sox3, Hes8, Hes9, Ngn1, Atoh1, Isl2, Pou4f1.2, Tlx1 and Gfi1a (this study); Sox2 synergises with Six1/Eya1 (*Ahmed et al., 2012b*, *2012a*); Sox2 directly activates Atoh1 (*Ahmed et al., 2012a*) and Ngn1 (*Cimadamore et al., 2011*); Atoh1 and Ngn1 indirectly repress each other (*Gowan et al., 2001*); Ngn1 indirectly represses Sox2 (*Evsen et al., 2013*); Ngn1 directly activates NeuroD1 (*Seo et al., 2007*); Atoh1 positively autoregulates (*Helms et al., 2000*); Atoh1 indirectly represses Sox2 (*Neves et al., 2012*) and activates Gfi1 (*Wallis et al., 2003*); NeuroD1 directly activates Pou4f1.2 (*Hutcheson and Vetter, 2001*) and Isl1 (*Lee et al., 1995*); Pou4f1.2 directly activates Gfi1 (*Hertzano et al., 2004*) and indirectly activates Tlx1 (*Hutcheson and Vetter, 2001*).

between activation of progenitor and differentiation genes. Additionally, direct binding of Six1 to regulatory regions of targets identified in this study should be confirmed by methods such as ChIP-Seq.

The analysis of Six1 and Eya1 presumptive direct target genes presented here establishes a GRN regulating the development of cranial vertebrate sensory organs and neurons from the PPE (*Figure 10*), and identifies a large number of novel putative direct target genes encoding a diverse array of proteins. Among these are many promising candidates potentially involved in mediating the effects of Six1 or Eya1 on proliferation, morphogenesis and cytodifferentiation in developing placodes. This makes our data an invaluable repository of information for designing further functional studies on early sensory development in vertebrates. Finally, while our study focussed on the role of Six1 and Eya1 during sensory development, cell proliferation, morphogenesis and cytodifferentiation are also known to be affected in human patients in which Six1 and Eya1 are dysregulated, leading to sensory deficits after *Six1* or *Eya1* loss of function mutations (*Kochhar et al., 2007*) or enhanced tumour progression after *Six1* or *Eya1* up-regulation (*Blevins et al., 2015*; *Liu et al., 2016*). This suggests that many target genes identified in our study may also be misregulated in these diseases, potentially opening up exciting new avenues for therapeutic intervention.

# Materials and methods

## Expression constructs and morpholinos

Capped RNAs of *Xenopus Six1-GR, Eya1-GR* and *Six1-EnR* were made by in vitro transcription using the mMessage mMachine SP6 kit (Ambion, Austin, Texas) from the following templates: pCS2$^+$-GR-myc-Six1, pCS2$^+$-GR-myc-Eya1$\alpha$ (*Schlosser et al., 2008*) and pCS2-EnR-Six1 (*Brugmann et al., 2004*).

Translation blocking morpholinos (MO) for Six1 (*Six1*-MO1: 5′-GGAAGGCAGCATAGACATGGC TCAG-3′; *Six1*-MO2: 5′-CGCACACGCAAACACATACACGGG-3′) and Eya1 (*Eya1*-MO1: 5′-TACTA TGTGGACTGGTTAGATCCTG-3′; *Eya1*-MO2: 5′-ATATTTGTTCTGTCAGTGGCAAGTC-3′) were previously described (*Brugmann et al., 2004*; *Schlosser et al., 2008*). An *Eya1*-MO carrying 5 mismatches was used as a control (*Eya1*-mmMO; mismatches shown in lower case: 5′-ATtTTaGTTCTGaCAGTGGgAAcTC-3′).

## Microinjection

*Six1-GR* (500 pg), *Eya1-GR,* (500 pg), *Six1-EnR* (100 pg) mRNAs and *Six1*-MO1+2 (2 ng), *Eya1*-MO1 +2 (2 ng), and *Eya1*-mismatch-MO (2 ng) were freshly prepared before each injection. *lacZ* (250 pg) or *myc-GFP* (125 pg) mRNAs were co-injected to mark the injected side. Embryos of *Xenopus laevis* were obtained by in vitro fertilisation, staged according to (*Nieuwkoop and Faber, 1967*) and injected unilaterally into two-cell blastomeres according to standard procedures (*Sive et al., 2000*). *Six1-EnR* was injected at the four-cell stage into single blastomeres that give rise to the dorsal ectoderm as previously described (*Brugmann et al., 2004*).

## Conditional overexpression of GR-fusion constructs and isolation of placodal RNA

To obtain RNA for RNA-Seq or qPCR, both blastomeres of two-cell stage embryos were injected with either 1) *Six1-GR* (500 pg) + *myc-GFP* (125 pg), 2) *Eya1-GR* (500 pg) + *myc-GFP* (125 pg), or 3) *Six1-GR* (500 pg) + *Eya1-GR* (500 pg) + *myc-GFP* (125 pg). Each of these treatment groups was allowed to develop to early neural plate stage before being sorted under a fluorescent microscope. The lateral part of the preplacodal region (LPR of *Ahrens and Schlosser, 2005*) was explanted from GFP positive embryos (~100 per biological replicate) in 1 × MBSH (*Sive et al., 2000*) supplemented with 2 mM CaCl$_2$, 25 mg/l gentamycine (Sigma, St Louis, Missouri), 400 mg/l penicillin (Sigma), and 400 mg/l streptomycin sulphate (Sigma). Explants were pre-treated with 0.1 × modified Barth's solution (MBS; *Sive et al., 2000*) with cycloheximide (CHX; final concentration 10 µg/ml) for 30 min at 25°C. After pre-treatment, 50% of the explants were transferred to 0.1 × MBS with CHX + dexamethasone (DEX; final concentration 10 µM) and incubated for 2 hr 30 at 25°C (*Figure 1*) when control embryos had reached stage 20. Explants were immediately homogenised in Trizol

(Invitrogen, Carlsbad, California) and total RNAs extracted. Isolated RNA was quality assayed in an Agilent 2100 Bioanalyzer and all samples used for sequencing had an RIN >7.0.

## RNA-sequencing, mapping and annotation

Libraries were prepared from 1 mg total RNAs and subjected to deep sequencing with Illumina Hi-Seq1000 at the EMBL Genecore facility. Paired-end (100 bp) sequence reads were quality-filtered using Trimmomatic (*Bolger et al., 2014*), and mapped to the *Xenopus laevis* genome (XL7.0) with Bowtie2 (version 2.2.5; *Langmead and Salzberg, 2012*) and Tophat2; (version 2.0.13; *Kim et al., 2013*). An average of 65 million reads (~80% of quality filtered reads) were mapped with 90% of reads properly paired in sequencing across treatment groups. Transcript models were assembled using Cufflinks2 (version 2.1.1; *Trapnell et al., 2012*), and differential expression was determined using Cuffdiff2 (version 2.1.1; *Trapnell et al., 2012*). Gene models were annotated against a combined *Xenopus* mRNA database (*X. laevis*: ftp://ftp.xenbase.org/pub/Genomics/Sequences/xlaevisMRNA.fasta; *X. tropicalis*: ftp://ftp.xenbase.org/pub/Genomics/Sequences/xtropMRNA.fasta) using blastn with an e-value cut-off of 1E-5. Using this approach we were able to annotate an average of 80% of mapped reads.

## Differential expression analysis for individual treatment groups

Initially, two samples of CHX- and CHX+DEX-treated explants were independently collected, sequenced and mapped for each treatment group (injection of Six1-GR, Eya1-GR or Six1-GR+Eya1-GR), and were specified as two biological replicates in Cuffdiff. To preclude the inclusion of genes affected by DEX treatment alone, we also analysed explants taken from un-injected embryos and treated as above (CHX vs. CHX+DEX). Two biological replicates of this control treatment group were included in sequencing. Presumptive direct targets of Six1, Eya1 or Six1+Eya1 were determined by comparing Six1-GR, Eya1-GR or Six1-GR+Eya1-GR-injected embryos treated with CHX (as controls) against CHX+DEX-treated samples. Genes were considered to be differentially expressed if (1) the FPKM (Fragments Per Kilobase of exon per Million fragments mapped) for that gene was >1 in the CHX+DEX treatment group, (2) the gene was at least two-fold up-/down-regulated after CHX+DEX treatment compared to CHX treatment, (3) there was at least a two-fold difference between the control (un-injected) and experimental (injected with either Six1-GR, Eya1-GR or Six1-GR+Eya1-GR) fold change (FC) values in response to DEX treatment. The Pearson correlation was high for each of the treatment groups (>0.9 for all pairwise comparisons), indicating the similarity of expression profiles between independently treated samples.

## Re-analysis of differential expression for combined treatment groups

As a second approach to finding genes that showed differential expression in response to DEX treatment, RNA-Seq data of several treatment groups were merged to add statistical power to the analysis. In one analysis, all replicates from our three different treatment groups were considered as equivalent to focus on genes with similar differential expression profiles across all treatment groups (comprising the Six1+Eya1$_m$ dataset with six replicates). In another analysis, all treatment groups involving *Six1* overexpression (i.e. injection of Six1-GR alone or Six1-GR+Eya1-GR: Six1$_m$ with 4 replicates) were treated as equivalent as were all treatment groups involving *Eya1* overexpression (Eya1-GR, Six1-GR+Eya1-GR: Eya1$_m$ with 4 replicates). This allowed us to focus on genes whose activation was limited by either Six1 or Eya1 levels. We considered a gene to be significantly differentially expressed if it passed Cuffdiff's statistical test (q < 0.05) in addition to meeting the criteria outlined above.

## Gene set enrichment analysis (GSEA) and Gene Ontology

*Xenopus* annotations were converted to their human orthologs from the Human Uniprot database, and functionally annotated using the online tools 'PantherDB' (*Mi et al., 2013*; http://pantherdb.org) and 'DAVID' (*Huang et al., 2009*; https://david.ncifcrf.gov). For GSEA of placodal transcriptomes after injection of Six1 and/or Eya1, the placodal transcriptome of un-injected, CHX treated placodal explants was specified as a background set, whereas GSEA of the transcriptome of untreated explants was conducted using the default 'human dataset' in DAVID as background. The enrichment score (E) for each group is reported as the geometric mean of the EASE scores (a

modified Fisher's exact score) that are associated with the enriched annotation terms belonging to that group (*Huang et al., 2007*).

## cDNA synthesis and qPCR

RNA was extracted from explants after CHX or CHX+DEX treatment as detailed above. cDNA was synthesised using the QuantiTect Reverse Transcription Kit (Qiagen, Hilden, Germany), using 500 ng total RNA according to the manufacturer's protocol. qPCR was performed using Taqman reagents on a StepOne Plus machine (Applied Biosystems, Foster City, California), using *Smn2* as a reference (*Dhorne-Pollet et al., 2013*; *Supplementary file 4*). qPCR was performed in triplicate and the entire experiment was repeated three times from independently prepared RNA. Relative Quantification (RQ) values and $\log_2$ fold change (FC) were averaged across biological replicates.

## Subcloning and gene synthesis

The full coding region of *Hes8, Crem, FosB, Tbx15, Atoh1* and *Isl2* was synthesised from transcript models from RNA-Seq data (KT722743; KT722744; KT722745; KT722746; KT722747; KT722748) by Genescript into the cloning vector pUC57 and subsequently sub-cloned into the expression vector pCS2$^+$ using the following restriction sites: *Hes8* and *Crem:* ClaI/EcoRI; *Atoh1:* XbaI; *Tbx15* and *FosB:* BamHI/EcoRI; *Isl2:* EcoRI/StuI. Primers with added ClaI and EcoRI sites (to the forward and reverse primers, respectively) were designed (*Supplementary file 4*) to amplify the entire coding region of *Tbx6*, which was then subcloned into pCS2$^+$ between the ClaI/EcoRI sites. *Znf214, Mab21l2-b* and *Pou3f2b* were ordered (pCMV-SPORT6, Fisher Scientific, Waltham, Massachusetts; Clone IDS: 5512398, 5515985 and 4203106).

Hes9 (pCR4-TOPO) was ordered from Source Bioscience (Clone accession: BC169570) and was subcloned into the EcoRI site of pCS2$^+$.

## In-situ-hybridisation

Embryos injected with *myc-GFP* were sorted under a fluorescent microscope and fixed using a standard protocol (*Sive et al., 2000*). *LacZ*-injected embryos were fixed and then stained with X-gal solution to reveal lacZ. Wholemount in-situ-hybridisation was carried out under high stringency conditions at 60°C as previously described (*Harland, 1991*) using digoxigenin-labelled antisense probes. Probes for *Six1* (*Pandur and Moody, 2000*), *N-tubulin* (*Oschwald et al., 1991 Sox2* (*De Robertis et al., 1997*), *Sox3* (*Penzel et al., 1997*), *Ripply3* (*Janesick et al., 2012*), *Hes2* (*Sölter et al., 2006*), *Sim1* (*Martin et al., 2007*), *Gbx2* (*von Bubnoff et al., 1996*), *Lhx5* (*Bachy et al., 2001*), *Sox21* (*Cunningham et al., 2008*), *Emx1.2* (*Green and Vetter, 2011*), *Pou4f1.2* (*Hutcheson and Vetter, 2001*), and *Tlx1* (*Patterson and Krieg, 1999*) were synthesised as previously described. Primers were designed with promoter sites added (forward, T7; reverse, SP6) for *Hes8, Hes9, Gfi1a, Tbx15, Ngn1, Pou4f1.2* and *Isl2* and were used to amplify a ~800 bp fragment from plasmid DNA (*Supplementary file 4*) which was then used as a template for probe synthesis using T7 RNA polymerase to make an antisense probe. pCMV-SPORT6 with *Znf214, Mab21l2-b* and *Pou3f2b* were linearised with SalI and antisense probes synthesised with T7. pCS2$^+$ vectors containing *Tbx6, FosB* and *Crem* were linearised with BamHI and transcribed with T7. pCS2$^+$ with *Atoh1* was linearised with NotI and transcribed with SP6.

## Vibratome sections and immunohistochemistry

In order to analyse the distribution of gene expression domains in finer detail, serial 40–50 µM vibratome sections were cut from selected embryos after wholemount in-situ hybridisation. Where staining with X-gal was insufficient to reveal the injected site, lacZ distribution was revealed immunohistochemically using a polyclonal rabbit anti-LacZ (MP Biomedicals Cappel, Santa Ana, California; Cat.: 55976; 1:1000) and an Alexa594-conjugated anti-rabbit antibody (1:1000).

## Availability of data and material

All sequencing data have been deposited in the NCBI BioProject database under BioProject PRJNA317049. All scripts used in analysis are available at https://github.com/nriddiford/Six1-Eya1-RNA-Seq.git.

## Acknowledgements

We thank Cathal Seoighe for advice regarding the experimental setup and bioinformatic analysis. Bruce Blumberg, André Brändli, Aldo Ciau-Uitz, Paul A Krieg, Roger Patient, Tomas Pieler, Sylvie Retaux, Elena Silva, and Monica Vetter kindly provided plasmids. We thank the GeneCore Sequencing Facility (EMBL, http://www.genecore.embl.de) and Vladimir Benes for RNA sequencing.

## Additional information

### Funding

| Funder | Grant reference number | Author |
| --- | --- | --- |
| Science Foundation Ireland | 11/RFP/EOB/3168 | Gerhard Schlosser |

The funders had no role in study design, data collection and interpretation, or the decision to submit the work for publication.

### Author contributions

NR, Carried out the experimental work and bioinformatic analysis, Wrote the manuscript; GS, Conceived the project, Assisted with several experiments, Wrote the manuscript

### Author ORCIDs

Nick Riddiford, http://orcid.org/0000-0002-4739-4233
Gerhard Schlosser, http://orcid.org/0000-0002-1300-1331

### Ethics

Animal experimentation: All animal experiments were performed in full accordance with Irish and European legislation, were approved by the NUI Galway Animal Care Research Ethics Committee (ACREC, 003/10) and were covered under the animal license (Cruelty to Animals Act, 1876) B100/4291 to G. Schlosser.

## Additional files

### Supplementary files

• Supplementary file 1. Top 1000 expressed genes (by FPKM) in the placodal transcriptome.

• Supplementary file 2. Tables of differentially expressed genes after overexpression of Six1-GR, Eya1-GR or Six1-GR+Eya1-GR: Individual treatment groups. Table 1: Genes with at least two-fold up-regulation after injection of Six1-GR and treatment with CHX + DEX. Table 2: Genes with at least two-fold up-regulation after injection of Eya1-GR and treatment with CHX + DEX. Table 3: Genes with at least two-fold up-regulation after injection of Six1-GR+Eya1-GR and treatment with CHX + DEX. Table 4: Genes with at least two-fold down-regulation after injection of Six1-GR and treatment with CHX + DEX. Table 5: Genes with at least two-fold down-regulation after injection of Eya1-GR and treatment with CHX + DEX. Table 6: Genes with at least two-fold down-regulation after injection of Six1-GR+Eya1-GR and treatment with CHX + DEX.

• Supplementary file 3. Tables of differentially expressed genes after overexpression of Six1-GR, Eya1-GR or Six1-GR+Eya1-GR: Merged treatment groups. Table 1: Genes with at least two-fold up-regulation after injection of Six1-GR or Six1-GR+Eya1-GR and treatment with CHX + DEX. Table 2: Genes with at least two-fold up-regulation after injection of Eya1-GR or Six1-GR+Eya1-GR and treatment with CHX + DEX. Table 3: Genes with at least two-fold up-regulation after injection of Six1-GR or Eya1-GR or Six1-GR+Eya1-GR and treatment with CHX + DEX. Table 4: Genes with at least two-fold down-regulation after injection of Six1-GR or Six1-GR+Eya1-GR and treatment with CHX + DEX. Table 5: Genes with at least two-fold down-regulation after injection of Eya1-GR or Six1-GR+Eya1-GR and treatment with CHX + DEX. Table 6: Genes with at least two-fold down-regulation after injection of Six1-GR or Eya1-GR or Six1-GR+Eya1-GR and treatment with CHX + DEX.

• Supplementary file 4. Primer sequences with modifications.

**Major datasets**

The following dataset was generated:

| Author(s) | Year | Dataset title | Dataset URL | Database, license, and accessibility information |
|---|---|---|---|---|
| Riddiford N, Schlosser G | 2016 | PRJNA317049 | http://www.ncbi.nlm.nih.gov/bioproject?LinkName=biosample_bioproject&from_uid=4640079 | Publicly available at the NCBI BioProject database (accession no: PRJNA317049) |

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
