## [Decision Letter]

Thank you for submitting your article "Dissecting the pre-placodal transcriptome to reveal direct targets of Six1 and Eya1 in cranial placodes" for consideration by *eLife*. Your article has been reviewed by two peer reviewers, and the evaluation has been overseen by K VijayRaghavan as the Senior Editor and Reviewing Editor. The following individuals involved in review of your submission have agreed to reveal their identity: Sally Moody (Reviewer #1) and Jean-Pierre Saint-Jennet (Reviewer #2)

The reviewers have discussed the reviews with one another and the Reviewing Editor has drafted this decision to help you prepare a revised submission.

Summary:

This is an interesting transcriptomic analysis of the direct targets of Six1 and Eya1 in the pre-placodal ectoderm (PPE). This is a very important analysis because both genes are critical for the development of cranial sensory organs and they are mutated in a human developmental defect, branchio-oto-renal syndrome. Although the role of Six1 and Eya1 in cranial placode development has been established for over 15 years, their direct transcriptional targets have not yet been analyzed in a comprehensive manner in a vertebrate model. This study uses techniques that have been very successful in *Xenopus* to detect direct transcriptional targets by expressing hormone inducible versions of Six1 and Eya1 in the presence of protein synthesis inhibitors. The authors have meticulously dissected out the precursor cells of the cranial placodes, the PPE, which provides important tissue specificity to the results. In addition, they make important comparisons between Six1 only, Eya1 only and Six1 + Eya1 targets. This is important because the literature shows in both fly and vertebrates that both proteins have alternate binding partners that can influence target gene specificity. This work is carefully carried out, includes appropriate controls and provides important new information for the community.

Some of the strengths/novelty of this manuscript are:

1) A comparison to a neural crest data set (Simoes-Costs 2014), which is interesting because it is a related but distinct tissue.

2) Inclusion of a Chx-treated only sample, which would identify genes that are expressed when repression is relieved.

3) qPCR validation of RNA-seq and excellent controls for the MO-knockdown experiments.

4) Confirmation of known Six1 targets based on the literature

5) Implication that Eya1 may not always act as a co-activator with Six1; this could be very important and explain some unexplained observations in the literature.

6) Solid evidence that Six1 works with other co-factors and Eya1 works with other partners in PPE development.

7) Potential role of Eya1 in regulating Hox genes

8) Comparison of the RNA-seq data to previously published microarray analysis.

Essential revisions: (These are all straightforward)

Our main concern with the manuscript is whether the authors have truly identified direct targets of Six1 and Eya1, as the title suggest. We are not sure based on the experimental approach used here, that the authors can make this claim. Direct targets cannot be defined just based on cycloheximide treatment experiments, especially since there is no indication of the efficiency of the treatment at blocking protein synthesis in this assay. Moreover, cycloheximide has been shown to alter gene expression on its own (see Sinner et al., 2004) – though the authors appear to have the appropriate controls. The identification and validation of Six1 binding sites in the regulatory regions of these targets would make for a much stronger case in favor of genuine direct targets. Until this is done, we suggest that the authors revise the wording throughout the manuscript by using terms such as "potential direct targets" or "presumptive direct targets". While it is very probable their list of genes contain direct targets, it seems unlikely that Six1 and Eya1 directly regulate the expression of several hundred genes in the PPE as stated at the beginning of the Discussion.

As a way of validating the approach and the quality of the data, the authors searched for established targets of Six1 in their data sets, however some of these genes (*Sox2* and Atoh1) are also upregulated in the Eya1 samples (Figure 3—figure supplement 1). What does it say about these targets and the quality of the data if they can be regulated independently of Six1? This should be discussed.

While this group has the demonstrated expertise to identify the various placodal domains at different stages of *Xenopus* development, it would be useful in some instances to confirm the placodal expression domain of these targets using well-characterized genes expressed in the same domain or in adjacent tissues, either using double ISH or single ISH in stage-matched embryos. For example, the expression pattern of Hes2 at the neurula stage appears to be largely confined to the neural crest territory, but Figure 4 indicate that it is expressed in the posterior placodal region.

---

## [Author Response]

*Essential revisions: (These are all straightforward)*

*Our main concern with the manuscript is whether the authors have truly identified direct targets of Six1 and Eya1, as the title suggest. We are not sure based on the experimental approach used here, that the authors can make this claim. Direct targets cannot be defined just based on cycloheximide treatment experiments, especially since there is no indication of the efficiency of the treatment at blocking protein synthesis in this assay. Moreover, cycloheximide has been shown to alter gene expression on its own (see Sinner et al., 2004) – though the authors appear to have the appropriate controls. The identification and validation of Six1 binding sites in the regulatory regions of these targets would make for a much stronger case in favor of genuine direct targets. Until this is done, we suggest that the authors revise the wording throughout the manuscript by using terms such as "potential direct targets" or "presumptive direct targets". While it is very probable their list of genes contain direct targets, it seems unlikely that Six1 and Eya1 directly regulate the expression of several hundred genes in the PPE as stated at the beginning of the Discussion.*

We have now worded the manuscript more carefully and use “presumptive direct targets” throughout.

*As a way of validating the approach and the quality of the data, the authors searched for established targets of Six1 in their data sets, however some of these genes (Sox2 and Atoh1) are also upregulated in the Eya1 samples (Figure 3—figure supplement 1). What does it say about these targets and the quality of the data if they can be regulated independently of Six1? This should be discussed.*

Since many Six1 target genes are coregulated by Eya1 this is actually to be expected. To clarify this issue we have now added the following sentences to the relevant section: “Moreover, *Atoh1, Sox2* and *MyoD1* were found both in our Six1+Eya1_i_ and Eya1_i_ datasets as expected based on the known coregulation of these Six1 target genes by Eya1 (Grifone et al., 2007; Ahmed et al., 2012 a; Schlosser et al., 2008). We suggest that overexpression of Eya1 alone may upregulate such genes in those parts of the ectoderm where Six1 is already expressed at high levels but Eya1 at relatively low levels in vivo.”

*While this group has the demonstrated expertise to identify the various placodal domains at different stages of Xenopus development, it would be useful in some instances to confirm the placodal expression domain of these targets using well-characterized genes expressed in the same domain or in adjacent tissues, either using double ISH or single ISH in stage-matched embryos. For example, the expression pattern of Hes2 at the neurula stage appears to be largely confined to the neural crest territory, but Figure 4 indicate that it is expressed in the posterior placodal region.*

Following the reviewer’s suggestions, we now have added a new figure (Figure 4—figure supplement 5) showing cross-sections through neural plate stage embryos expressing Hes2, Hes8, Hes9 and Pou4f1.2 in comparison to cross-sections from embryos after double in situ for Six1/Sox3 and Six1/FoxD3 (the latter taken from the previous study of Schlosser and Ahrens, 2004). This shows that Hes2 as well as the other genes are predominantly expressed in the PPE and not the neural crest at neural plate stages (even though some overlap with the lateral part of the neural crest cannot be ruled out based on our data). We also have added images of Six1 expression in stage-matched embryos as an additional row to Figure 4—figure supplement 1-4 to facilitate comparison of expression domains.